# A Fair Federated Learning Method for Handling Client Participation Probability Inconsistencies in Heterogeneous Environments

**Siyuan Wu**[1]**, Yongzhe Jia**[1]**, Haolong Xiang**[2]**, Xiaolong Xu**[2]**,**
**Xuyun Zhang**[3]**, Lianyong Qi**[4]**, Wanchun Dou**[1,*]
[1]State Key Laboratory for Novel Software Technology, Nanjing University, China
[2]School of Computer and Software,
Nanjing University of Information Science and Technology, China
[3]School of Computing, Macquarie University, Australia
[4]College of Computer Science and Technology, China University of Petroleum (East China), China

## Abstract

Federated learning (FL) is a distributed machine learning paradigm that enables multiple clients to collaboratively train a shared model without exposing their raw data. However, existing FL research has primarily focused on optimizing learning performance based on the assumption of uniform client participation, with few studies delving into performance fairness under inconsistent client participation, particularly in model-heterogeneous FL environments. In view of this challenge, we propose **PHP-FL**, a novel model-heterogeneous FL method that explicitly addresses scenarios with varying client participation probabilities to enhance both model accuracy and performance fairness. Specifically, we introduce a Dual-End Aligned ensemble Learning (DEAL) module, where small auxiliary models on clients are used for dual-end knowledge alignment and local ensemble learning, effectively tackling model heterogeneity without a public dataset. Furthermore, to mitigate update conflicts caused by inconsistent participation probabilities, we propose an Importance-driven Selective Parameter Update (ISPU) module, which accurately updates critical local parameters based on training progress. Finally, we implement PHP-FL on a lightweight FL platform with heterogeneous clients across three different client participation patterns. Extensive experiments under heterogeneous settings and diverse client participation patterns demonstrate that PHP-FL achieves state-of-the-art performance in both accuracy and fairness. Our code is available at: `https://github.com/Siyuan01/PHP-FL-main`.

## 1 Introduction

Federated Learning (FL) has emerged as a promising paradigm for enabling decentralized model training across multiple clients without directly sharing their private data [1, 2]. By collaboratively learning a global model while keeping data localized, FL offers strong privacy guarantees and broad applicability across various domains such as mobile devices, healthcare, and finance [3–5].

Despite significant progress, traditional FL methods still face two critical challenges: **I) Model heterogeneity.** Traditional FL assumes that all clients share an identical local model architecture, which is often impractical in real-world due to the diversity in client capabilities. To address this, Model-Heterogeneous Federated Learning (MH-FL) has emerged as a promising research paradigm [2, 6–9], which allows each client to maintain a personalized model tailored to its own resource constraints

---

*Corresponding author: Wanchun Dou (douwc@nju.edu.cn)

39th Conference on Neural Information Processing Systems (NeurIPS 2025).

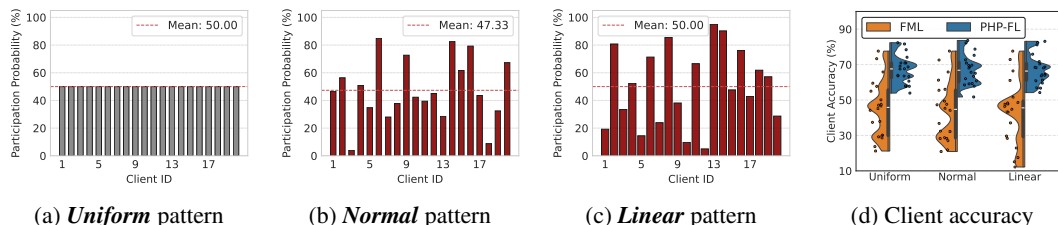

|  (a) *Uniform* pattern | (b) *Normal* pattern | (c) *Linear* pattern | (d) Client accuracy |

Figure 1: Left three: three client participation probability patterns (average value is approximately 50%). Right: client accuracy under patterns (a)-(c). Refer to Section 5.1 for experimental details.

or task requirements. However, most existing MH-FL methods [10–13] primarily aim to ensure compatibility between diverse model architectures, yet overlook the fairness issues posed arising from inconsistent client participation probabilities in practical deployments. This oversight can result in biased models that favor frequently participating clients, ultimately compromising the robustness and fairness of the FL system. **II) Unfairness caused by inconsistent client participation.** Most existing FL research [1, 9, 12–16] implicitly assume a uniform client participation pattern, where all clients are equally likely to participate in each training round, as illustrated in Figure 1a. In practical deployments, clients often face heterogeneous conditions, such as intermittent connectivity, fluctuating network bandwidth, and network coverage of base stations [17–20]. These internal and external factors lead to client unavailability and result in non-uniform participation probabilities, as exemplified in Figure 1b and 1c, which may critically degrade both the overall accuracy and fairness of the FL system. Figure 1d compares the final client accuracy distributions between FML [21] (a representative MH-FL method) and our method PHP-FL under three participation patterns. While FML suffers significant performance degradation in the *normal* and *linear* patterns compared to the *uniform* participation, PHP-FL maintains stable accuracy with only marginal drops. Furthermore, PHP-FL demonstrates tighter accuracy distributions, indicating better performance fairness. Although some studies [22–25] improve overall system performance by proactively selecting high-availability clients and discarding less efficient ones, such strategies often compromise fairness across clients. Furthermore, numerous fair federated learning methods aim to enhance performance fairness through personalized model [14, 26] or weight recalibration [27, 28]. Nevertheless, they typically operate under the assumption of uniform client participation, failing to address the challenge of inconsistent client availability. Despite its importance, this issue has received limited attention in the literature, particularly in the context of MH-FL, where the interplay between model heterogeneity and participation imbalance exacerbates the learning challenge.

To address these challenges, we propose PHP-FL, a fair federated learning method designed to address scenarios with varying client participation probabilities in model-heterogeneous environments. Specifically, to effectively tackle model heterogeneity without relying on public datasets, we introduce a Dual-End Aligned ensemble Learning (DEAL) module, which leverages lightweight auxiliary models on clients to align heterogeneous local models and enable ensemble learning to improve the performance of local tasks. Furthermore, to mitigate the adverse effects of update conflicts that caused by inconsistent client participation probabilities, we propose a Importance-driven Selective Parameter Update (ISPU) module. The ISPU module adaptively updates only the most critical task-relevant parameters based on training progress, allowing clients with different participation frequencies to selectively absorb varying ratios of global knowledge. This design helps reduce gradient conflicts and enhancing fairness. Our main contributions are as follows:

- To the best of our knowledge, this is the first work to explicitly address performance unfairness caused by inconsistent client participation probabilities in practical FL systems with heterogeneous local models.

- We propose PHP-FL, a novel model-heterogeneous federated learning method designed to address inconsistent client participation probabilities, aiming to jointly improve both overall accuracy and performance fairness across clients.

- We evaluate PHP-FL through extensive experiments on a lightweight FL platform simulating multiple realistic participation patterns. Empirical results on the Fashion-MNIST and CIFAR-10 datasets demonstrate its state-of-the-art performance, while the ablation study further validates the effectiveness of each proposed module.

## 2 Related Works

**Model-Heterogeneous Federated Learning.** Model-Heterogeneous Federated Learning (MH-FL) has emerged as a promising research direction [11, 6, 8, 9, 2, 13]. Existing work in this area can be broadly categorized into *knowledge distillation*-based (KD) methods, *representation alignment* methods, and *partial model sharing* methods. KD is one of the most widely adopted techniques in MH-FL. The studies in [29–31] enable clients with different architectures to distill knowledge through a shared or public dataset. However, such reliance on public data limits applicability in privacy-sensitive settings. Another popular line of work, like [15, 12, 16], focuses on representation alignment, which aligns feature representations or prototypes rather than raw model parameters, allowing clients to maintain model diversity while contributing to a shared learning objective. Furthermore, some studies [7, 32–34] adopt partial model sharing strategies, where clients share only specific model components (*e.g.*, a shared backbone or predictor) while keeping the other parts distinct, thereby enabling partial compatibility across models. However, few MH-FL methods explicitly address fairness for clients under inconsistent participation probabilities, a critical requirement for equitable and robust deployment.

**Fairness in Federated Learning.** Existing research on fairness in federated learning has primarily focused on three key dimensions: (1) *Contribution Fairness* [4, 35, 36], which involves evaluating each client's contribution to the global model to guide equitable benefit distribution, often using techniques like Shapley value or influence functions [37, 38]; (2) *Model Fairness* [39, 40], which addresses inherent biases in model predictions concerning sensitive attributes, thereby promoting fairness at the prediction level; and (3) *Performance Fairness* [14, 20, 27, 41, 28], which aims to ensure uniform model performance across clients, typically by minimizing the variance or standard deviation of test accuracy among clients. As prior studies [42, 43] demonstrate, these fairness metrics often conflict, making it infeasible for a model to simultaneously achieve optimal performance across all dimensions. Our work, therefore, specifically targets performance fairness, aiming to ensure uniform performance across clients while concurrently optimizing overall performance under inconsistent participation probabilities in MH-FL. While another line of research [20, 44, 45] addresses client unavailability by primarily focusing on maintaining the average performance across clients, they do not explicitly ensure performance-level fairness. Furthermore, these methods are typically designed for homogeneous settings and face significant challenges when generalizing to heterogeneous federated learning environments, where client capabilities vary substantially.

## 3 Preliminaries

**The Global Objective of Federated Learning.** Following typical federated learning [1, 26] settings, we consider a set of $K$ clients (index by $i$) with local datasets $\{D_1, D_2, ..., D_K\}$, where $D_i = \{(x_j, y_j)\}_{j=1}^{n_i}$ and $n_i = |D_i|$. In a heterogeneous FL environment, each client $i$ maintains a unique model $\boldsymbol{w}_i$, parameterized by $\theta_i \in \mathbb{R}^{d_i}$, with dimensional heterogeneity ($d_i \neq d_j$) arising from hardware constraints (*e.g.*, compute/memory limits) or personalized model specialization. This implies $\dim(\theta_i) \neq \dim(\theta_j), \exists i \neq j \in [K]$. The global objective function can be expressed as:

$$\min_{\{\boldsymbol{w}_i\}_{i=1}^K} F(\{\boldsymbol{w}_i\})_{i=1}^K = \sum_{i=1}^K p_i F_i(\boldsymbol{w}_i), \quad \sum_{i=1}^K p_i = 1, \tag{1}$$

where $p_i$ is the weight of client $i$, $\{\boldsymbol{w}_i\}_{i=1}^K$ represents the set of the client's local models, and $F_i(\boldsymbol{w}_i) = \frac{1}{n_i} \sum_{j=1}^{n_i} \mathcal{L}(\boldsymbol{w}_i(\theta_i; x_j), y_j)$ is the local objective for client $i$ with loss function $\mathcal{L}$.

**Inconsistent Client Participation Probability.** To simulate realistic client availability in federated learning, we consider three types of client participation probability patterns. Let $p_{i,t}$ denote the probability that client $i$ actively participates in communication round $t$:

**Definition 1** *(Uniform Pattern) All clients share an identical and fixed probability $a \in (0, 1]$ of participating in each round , i.e., $p_{i,t} = a, \quad \forall i \in \{1, 2, \ldots, n\}, \forall t$.*

**Definition 2** *(Normal Pattern) The participation probabilities are drawn from a truncated normal distribution to simulate natural heterogeneity: $p_{i,t} \sim \mathcal{N}(\mu, \sigma)$, and $p_{i,t}$ is clipped to $(0, 1]$.*

**Definition 3** *(Linear Pattern) Client participation probabilities are distributed according to an increasing arithmetic sequence: $p_{i,t} = a + (i-1)d, \quad i = 1, 2, \ldots, n, \ \forall t$, where $a$ is the first term and $d$ is the common difference. In this pattern, the initial sequence satisfies $0 < p_{1,t} < p_{2,t} < \cdots < p_{K,t} \leq 1$ for round $t$. This sequence $\{p_{i,t}\}_{i=1}^{K}$ is then randomly shuffled prior to use to eliminate any inherent ordering bias among clients. It models systematic heterogeneity such as time-varying connectivity or device capacity.*

Note that $p_{i,t}$ is independent of the history and other clients. The three distinct patterns considered enable a comprehensive analysis of how heterogeneous client participation affects both fairness and overall performance in federated learning.

**Design Goals.** In this paper, we aim to design a MH-FL method under inconsistent client participation probabilities that not only optimizes the *average performance* across all clients but also enhances *performance fairness*. Formally, let $a_i (i = 1, \ldots, K)$ represent the test accuracy on the $i$-th client's local test dataset. The Accuracy Metric (AM) is defined as: $AM = \frac{1}{K} \sum_{k=1}^{K} a_k$. The Fairness Metric (FM) is defined as: $FM = \text{Std}(a_1, \ldots, a_K)$, where $\text{Std}(\cdot)$ denotes the standard deviation. To this end, our approach seeks to maximize the average local accuracy (AM) while minimizing the performance disparity (FM), ensuring both high overall performance and fair performance distribution across clients.

## 4 Methodology

### 4.1 Overview of PHP-FL

Our method consists of two key modules: dual-end aligned ensemble learning (DEAL in Section 4.2), and importance-driven selective parameter update (ISPU in Section 4.3). The DEAL module employs a small homogeneous auxiliary model to perform bidirectional representation-logit alignment between local and auxiliary models, which resolves model heterogeneity and enhances overall performance via ensemble learning. To handle inconsistent client participation, the ISPU module selectively updates task-relevant critical parameters using an importance-based masking mechanism. This approach applies larger updates to stragglers to accelerate overall convergence, while reducing updates for frequent participants to prevent adverse effects from the stragglers. This ensures efficient knowledge fusion and fair parameter evolution across clients.

As shown in Figure 2, the training process in each communication round $t$ of PHP-FL can be summarized as follows:[2] ❶ The server first computes the client-specific auxiliary model $\mathcal{G}^{t-1} \odot M_i^h$ for the active client $i \in \mathcal{A}_t$ by pruning non-essential parameters using historical binary mask $M_i^h$. ❷ Then active clients initialize the personalized auxiliary models $\hat{g}_i^{t-1}$. ❸ PHP-FL decomposes both the local model $\boldsymbol{w}_i$ and the auxiliary model $\boldsymbol{g}_i$ into a *backbone* and a *predictor*, which are used for representation extraction and soft prediction, respectively. At the beginning of local training, the DEAL module first optimizes the ensemble weights $\boldsymbol{\lambda}_i$ on the adaptation set $\mathcal{D}_i^a$, while $\boldsymbol{w}_i^t$ and $\hat{\boldsymbol{g}}_i^{t-1}$ are frozen. ❹ Local training then proceeds using the customized loss $\mathcal{L}_{\boldsymbol{w}}$, which comprises two components: (1) The *dual-end alignment loss* $\mathcal{L}_{\text{DEAL}}$, enabling bidirectional knowledge alignment through data-free distillation and representation matching. (2) The *ensemble learning loss* $\mathcal{L}_{\text{ENS}}$, which further enhances overall model performance. ❺ Following the local training, the ISPU module calculates an update ratio $\alpha_i^t$ based on the client's total training rounds. It then estimates the top-$\alpha_i^t$ important parameters of $\boldsymbol{g}_i^t$ using the $\ell_1$-norm and generates a binary mask $M_i^t$. Finally, each active client uploads both its updated local auxiliary model $\boldsymbol{g}_i^t$ and the binary mask $M_i^t$ to the server. ❻ The server updates the historical mask $M_i^h \in M_{\text{hist}}$ for each active client $i$ by replacing its entry with the newly received $M_i^t$. Then the received auxiliary models are aggregated via a simple averaging technique to obtain the global auxiliary model $\mathcal{G}^t$ for round $t + 1$.

### 4.2 Dual-End Aligned Ensemble Learning

To address system heterogeneity without relying on public datasets, previous works such as [12, 15] decompose the model $w$[3] (parameterized by $\theta$) into a *backbone* $\boldsymbol{w}_b$ and a *predictor* $\boldsymbol{w}_p$, and perform

---

[2]Algorithm 1 in Appendix A describes the PHP-FL algorithm.

[3]We omit the client index $i$ and the communication round index $t$ for notation simplicity.

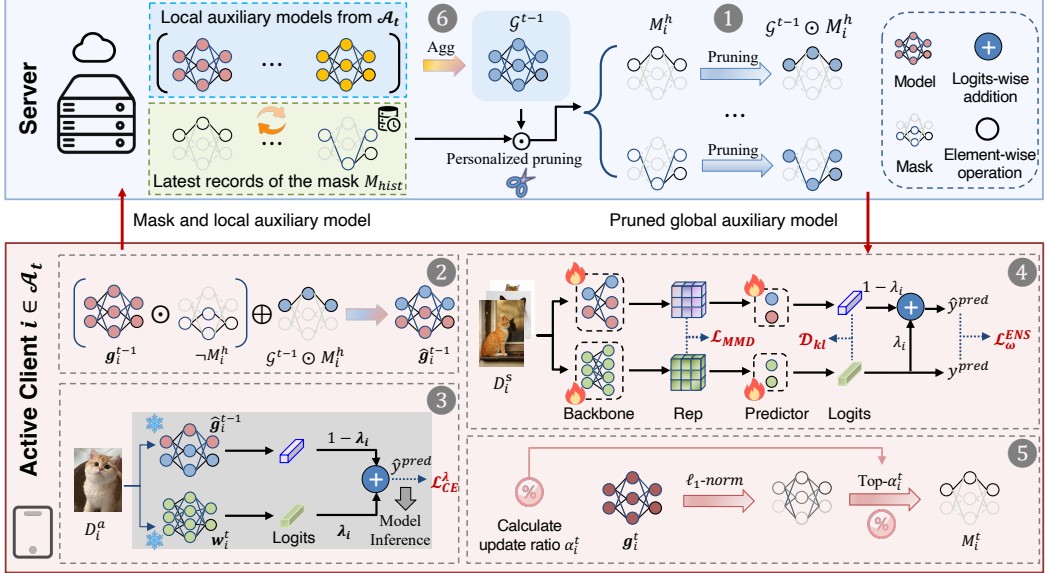

Figure 2: The overview of PHP-FL.

aggregation based on intermediate representations $z = \boldsymbol{w}_b(\theta_b; x_j)$ produced by the backbone. However, it's challenging to classify representations generated by heterogeneous backbones for local predictors. Inspired by the spirit of the mutual learning, some works [46, 47] leverage logits-level knowledge distillation, where each client co-trains a heterogeneous local model $\boldsymbol{w}$ and a lightweight homogeneous global model $\boldsymbol{g}$ (parameterized by $\phi$) by aligning only the final outputs of the predictors. Unfortunately, this limited alignment fails to facilitate meaningful knowledge transfer to the backbone component, resulting in suboptimal representation learning. To address these limitations, we propose a DEAL module. In DEAL, both the backbone and predictor components between local and auxiliary models are explicitly aligned. This dual-end alignment ensures effective and rapid fusion of local and global knowledge. To this end, we design the following loss function:

$$\mathcal{L}^{\boldsymbol{w}}_{DEAL} = \frac{1}{|D_i|} \sum_{j \in D_i} [\underbrace{\mathcal{L}_{\text{MMD}}(\boldsymbol{w}_b(\theta_b; x_j), \boldsymbol{g}_b(\phi_b; x_j))}_{\text{Backbone Alignment}} + \underbrace{\mathcal{D}_{\text{KL}}\left(\boldsymbol{w}(\theta; x_j) \| \boldsymbol{g}(\phi; x_j)\right)}_{\text{Predictor Alignment}}], \quad (2)$$

where $\mathcal{D}_{\text{KL}}$ is Kullback-Leibler (KL) divergence. The Maximum Mean Discrepancy (MMD) loss $\mathcal{L}_{\text{MMD}}$ between two sets of representations $\mathbf{z}_1 \in \mathbb{R}^{n \times d_1}$ and $\mathbf{z}_2 \in \mathbb{R}^{m \times d_2}$ using a Gaussian radial basis function (RBF) kernel is computed as:

$$\begin{aligned}
\mathcal{L}_{\text{MMD}}(\mathbf{z}_1, \mathbf{z}_2) = \frac{1}{n^2} \sum_{i,j=1}^{n} k(f(\mathbf{z}_1^{(i)}), f(\mathbf{z}_1^{(j)})) + \frac{1}{m^2} \sum_{i,j=1}^{m} k(h(\mathbf{z}_2^{(i)}), h(\mathbf{z}_2^{(j)})) \\
- \frac{2}{nm} \sum_{i=1}^{n} \sum_{j=1}^{m} k(f(\mathbf{z}_1^{(i)}), h(\mathbf{z}_2^{(j)})),
\end{aligned} \quad (3)$$

where $f : \mathbb{R}^{d_1} \to \mathbb{R}^d$ and $h : \mathbb{R}^{d_2} \to \mathbb{R}^d$ represent customizable projection functions designed to standardize feature dimensions, enabling cross-architecture feature alignment between the local model $\boldsymbol{w}$ and global model $\boldsymbol{g}$ when their structures are heterogeneous. The Gaussian RBF kernel $k(\mathbf{z}, \mathbf{z}') = \exp\left(-\gamma \|\mathbf{z} - \mathbf{z}'\|^2\right)$ measures similarity between representations, with $\gamma = 1/(2\sigma^2)$ controlling the kernel bandwidth. This non-parametric metric effectively captures the distance between the distributions of $\mathbf{z}_1$ and $\mathbf{z}_2$ in the Reproducing Kernel Hilbert space (RKHS). MMD is able to effectively align feature distributions by comparing global statistics via kernel-based embeddings, handles non-IID data robustly, and enables stable optimization [48].

Next, to fully leverage the classification capabilities of both local models and global auxiliary models, we adopt model ensembling [49, 47, 13] to enhance performance on local tasks:

$$\mathcal{L}_{ENS}^{\boldsymbol{w}} = \frac{1}{|D_i|} \sum_{j \in D_i} [\mathcal{L}_{\text{CE}}(\boldsymbol{w}(\theta; x_j), y_j) + \mathcal{L}_{\text{CE}}(\lambda \boldsymbol{w}(\theta; x_j) + (1 - \lambda)\boldsymbol{g}(\phi; x_j), y_j)], \quad (4)$$

where $\mathcal{L}_{\text{CE}}$ denotes the cross-entropy loss between the predicted label and the ground-truth label. Furthermore, to address the challenges posed by the potential heterogeneous system capabilities between each local model $\boldsymbol{w}$ and the global model $\boldsymbol{g}$, we set $\boldsymbol{\lambda}$ as a trainable parameter for each client and randomly hold out a tiny *adaptability set* $D_i^a$ from the training set $D_i$ (*e.g.*, 10%) for its optimization at the beginning of every communication round. The remaining set on each client for training is denoted as the *study set* $D_i^s$. This round-wise resampling of the adaptability set guarantees that the ensemble weight $\boldsymbol{\lambda}_i$ is continuously optimized on fresh and unbiased data, thereby effectively mitigating overfitting risks. The learning process of $\boldsymbol{\lambda}$ on each client is:

$$\boldsymbol{\lambda}^t \leftarrow \boldsymbol{\lambda}^{t-1} - \eta_{\boldsymbol{\lambda}} \frac{1}{|D_i^a|} \sum_{j \in D_i^a} \nabla_{\boldsymbol{\lambda}^{t-1}} \mathbb{E}_{(x_j, y_j) \sim D_i^a} \mathcal{L}_{\text{CE}}(\boldsymbol{\lambda}^{t-1} \boldsymbol{w}(\theta; x_j) + (1 - \boldsymbol{\lambda}^{t-1})\boldsymbol{g}(\phi; x_j), y_j), \quad (5)$$

where $\eta_{\boldsymbol{\lambda}}$ is the learning rate for $\boldsymbol{\lambda}$. Finally, the total training objective of the local model $\boldsymbol{w}$ combines the dual-end alignment loss and the ensemble learning loss:

$$\mathcal{L}_{\boldsymbol{w}} = \mathcal{L}_{DEAL}^{\boldsymbol{w}} + \mathcal{L}_{ENS}^{\boldsymbol{w}}, \quad (6)$$

For symmetry, an analogous loss $\mathcal{L}_{\boldsymbol{g}}$ is also computed but omitted here for brevity, as it follows the same formulation with reversed inputs. The total losses $\mathcal{L}_{\boldsymbol{g}}$ and $\mathcal{L}_{\boldsymbol{w}}$ are used to simultaneously update the homogeneous auxiliary model and the heterogeneous client local model, with learning rates $\eta_{\boldsymbol{g}}$ and $\eta_{\boldsymbol{w}}$, respectively, as follows:

$$\boldsymbol{w}^t \leftarrow \boldsymbol{w}^{t-1} - \eta_{\boldsymbol{w}} \frac{1}{|D_i^s|} \sum_{j \in D_i^s} \nabla_{\boldsymbol{w}^{t-1}} \mathbb{E}_{(x_j, y_j) \sim D_i^s} \mathcal{L}_{\boldsymbol{w}}(\boldsymbol{w}^{t-1}, \boldsymbol{g}^{t-1}, \boldsymbol{\lambda}^t, x_j, y_j),$$

$$\boldsymbol{g}^t \leftarrow \boldsymbol{g}^{t-1} - \eta_{\boldsymbol{g}} \frac{1}{|D_i^s|} \sum_{j \in D_i^s} \nabla_{\boldsymbol{g}^{t-1}} \mathbb{E}_{(x_j, y_j) \sim D_i^s} \mathcal{L}_{\boldsymbol{g}}(\boldsymbol{g}^{t-1}, \boldsymbol{w}^{t-1}, \boldsymbol{\lambda}^t, x_j, y_j). \quad (7)$$

During the inference stage, clients use the weighted model ensemble for prediction:

$$\hat{y}_j^{pred} = \arg\max(\boldsymbol{\lambda}\boldsymbol{w}(\theta; x_j) + (1 - \boldsymbol{\lambda})\boldsymbol{g}(\phi; x_j)). \quad (8)$$

This adaptive weighting mechanism automatically balances the contributions of both models based on their current performance. It is particularly effective under system heterogeneity, where devices may have varying computational capabilities.

### 4.3 Importance-Driven Selective Parameter Update

To enable straggling clients to quickly catch up upon rejoining training while preserving the learning momentum of more active clients, we propose a novel selective parameter update module, ISPU, which selectively updates task-relevant parameters and suppresses noisy or redundant updates. Instead of directly overwriting the local auxiliary model $\boldsymbol{g}_i^{t-1}$ with the global model $\mathcal{G}^{t-1}$, we perform a model fusion by identifying the most significant parameters in $\boldsymbol{g}_i^{t-1}$. The update ratio $\alpha_i^t$ is adaptively determined by a sigmoid-based scheduling function according to the training progress:

$$\alpha_i^t = \frac{1}{1 + \exp\left(\delta \cdot \left(\frac{N_i(t)}{t+1} - 0.5\right)\right)} \cdot \tau \quad (9)$$

where $N_i(t) = \sum_{r=1}^{t} \mathbb{1}\{i \in \mathcal{A}_r\}$ denotes the cumulative number of rounds in which client $i$ has participated up to round $t$ and $\mathbb{1}\{\cdot\}$ is the indicator function. Here, $\tau \in (0, 1]$ represents the pruning threshold and $\delta$ is a tunable sharpness hyperparameter. We then apply a binary mask on the parameters of $\boldsymbol{g}_i^{t-1}$ to retain only the critical parameters and replace them with the corresponding parameters from the global model $\mathcal{G}^{t-1}$. Common pruning metrics include the $\ell$-norm [50, 3], Fisher Information Matrix (FIM) [51, 52], and sensitivity-based measures [53, 54]. Specifically, we adopt

the $\ell_1$-norm to evaluate the importance of parameters, which has been proven to be an effective technique for assessing parameter significance [55, 3]. Compared to other metrics, this formulation better captures the importance of task-relevant parameters. After local training, the binary mask $M_i^t$ is constructed to update only the top-$\alpha_i^t$ important parameters of $\boldsymbol{g}_i^t$ in the next participation round:

$$M_{i,d}^t = \begin{cases} 1, & \text{if } d\text{-th parameter } \in \text{top-}\alpha_i^t \text{ largest of } \boldsymbol{g}_i^t \\ 0, & \text{otherwise} \end{cases} \tag{10}$$

This parameter-wise filtering mechanism helps **frequently active clients** preserve their learned knowledge while allowing **infrequent clients** to assimilate global updates more effectively, enabling rapid catch-up and mitigating knowledge drift. Specifically, at the beginning of round $t$, the local auxiliary model $\boldsymbol{g}_i^{t-1}$ of each active client $i \in \mathcal{A}_t$ is updated as follows:

$$\hat{\boldsymbol{g}}_i^{t-1} = \boldsymbol{g}_i^{t-1} \odot \neg M_i^h + \mathcal{G}^{t-1} \odot M_i^h, \tag{11}$$

where $M_i^h$ is the mask matrix obtained by client $i$ from its most recent training round and $\neg M$ denotes the bit-wise inverse of the mask $M$. This approach ensures clients preserve critical knowledge via high-importance parameters, filters out conflicts from heterogeneous data and inconsistent training progress.

## 5 Experiments

### 5.1 Experiments Details

**Datasets.** We evaluate our proposed PHP-FL on two standard image classification benchmarks: Fashion-MNIST[4] [56] and CIFAR-10[5] [57]. For both datasets, we adopt a 4:1 ratio to split samples into training and test sets. Following previous studies [12, 58, 59], we simulate heterogeneous data distributions by allocating class $j$ proportions to each client $k$ according to a Dirichlet distribution ($p_{j,k} \sim \text{Dir}(\beta)$), where a smaller $\beta$ implies more extreme data heterogeneity across clients. We adopt $\beta = 0.1$ for Fashion-MNIST and $\beta = 0.5$ for CIFAR-10, respectively. Notably, each client's local training and test sets share the same distribution.

**Models.** Our experimental setup employs four heterogeneous model architectures: (1) GoogleNet [60], (2) DenseNet-121 [61], (3) EfficientNet-B1 [62], and (4) ResNet-18 [63]. Each client is assigned one of these models based on its identifier $i$, following a round-robin strategy where client $i$ receives the model corresponding to $i \bmod 4$. Comprehensive comparative results of homogeneous model architectures are provided in Appendix C.2.

**Comparison Baselines.** In the heterogeneous model experiments, we comprehensively evaluate our method against six state-of-the-art heterogeneous federated learning algorithms without relying on public data, including FML [21], FedGen [10], FedKD [11], FedAPEN [47], FedTGP [12], FedMRL [13]. In addition, we also compare a standalone baseline where clients train locally without any aggregation or communication.

**Implementation Details.** Our experimental framework is built on a lightweight MH-FL platform HtFLlib [64] using PyTorch 2.2.2 [65] with NVIDIA RTX 3090 GPU. We employ SGD as our optimizer with a learning rate of 0.001 and a local batch size of 64. The global training process consists of 100 communication rounds, with a total of 20 clients. During each federated training round, clients perform 5 local epochs of training. In each round, client participation follows the three patterns introduced in Section 3. We repeated all experiments three times with different random seeds and present the averaged results. More details are provided in Appendix B.

**Evaluation Metrics.** As defined in Section 3, we evaluate the performance using the average Top-1 accuracy (AM) across all clients. In evaluating the fairness of the clients, we adopt the standard deviation (FM) of client accuracy when the Top-1 test accuracy is achieved. In PHP-FL, the test accuracy and fairness for each client are derived from the ensemble output of its local and auxiliary models, as computed by Eq. 8.

---

[4]`https://github.com/zalandoresearch/fashion-mnist?tab=readme-ov-file`
[5]`https://www.cs.toronto.edu/~kriz/cifar.html`

Table 1: Comparison with the state-of-the-art methods on Fashion-MNIST in the heterogeneous setting. Best in **bold** and second with underline. ↑ indicates improved accuracy (%) and ↓ indicates improved standard deviation (%) of accuracy compared with the best baseline, respectively.

| Methods | *Uniform* $[a = 0.5]$ | | *Normal* $[\mu = 0.5, \sigma = 0.2]$ | | *Linear* $\left[a = 0.05, d = \frac{K-2}{K(K-1)}\right]$ | |
|---|---|---|---|---|---|---|
| | AM (%) ↑ | FM (%) ↓ | AM (%) ↑ | FM (%) ↓ | AM (%) ↑ | FM (%) ↓ |
| Standalone | 95.88±0.19 | 9.26±1.37 | 96.89±0.19 | 9.37±1.22 | 95.77±0.05 | 9.86±0.53 |
| FML [arXiv20] | 89.09±0.66 | 23.13±1.18 | 88.71±0.23 | 23.32±1.11 | 88.15±0.73 | 25.21±2.76 |
| FedGen [ICML21] | 93.97±1.13 | 22.78±1.46 | 93.81±0.99 | 23.25±1.52 | 93.72±0.93 | 23.80±1.90 |
| FedKD [NC22] | 95.67±0.31 | 9.20±0.92 | 95.65±0.30 | 9.02±1.16 | 95.57±0.18 | 9.30±0.78 |
| FedAPEN [KDD23] | 96.79±0.06 | 6.74±0.26 | 96.79±0.07 | 6.50±0.60 | 96.73±0.06 | 6.89±0.08 |
| FedTGP [AAAI24] | 94.35±2.57 | 10.94±2.47 | 94.06±2.38 | 11.32±2.25 | 94.21±2.48 | 11.50±2.18 |
| FedMRL [NIPS24] | 96.06±0.45 | 9.07±1.65 | 96.05±0.43 | 9.25±1.39 | 95.78±0.09 | 9.81±0.64 |
| **PHP-FL (Ours)** | **97.64±0.04** | **4.15±0.18** | **97.59±0.04** | **4.24±0.05** | **97.58±0.03** | **4.29±0.04** |
| | ↑ **0.85** | ↓ **2.59** | ↑ **0.80** | ↓ **2.26** | ↑ **0.95** | ↓ **2.60** |

Table 2: Comparison with the state-of-the-art methods on CIFAR-10 in the heterogeneous setting. Best in **bold** and second with underline. ↑ indicates improved accuracy (%) and ↓ indicates improved standard deviation (%) of accuracy compared with the best baseline, respectively.

| Methods | *Uniform* $[a = 0.5]$ | | *Normal* $[\mu = 0.5, \sigma = 0.2]$ | | *Linear* $\left[a = 0.05, d = \frac{K-2}{K(K-1)}\right]$ | |
|---|---|---|---|---|---|---|
| | AM (%) ↑ | FM (%) ↓ | AM (%) ↑ | FM (%) ↓ | AM (%) ↑ | FM (%) ↓ |
| Standalone | 56.66±0.32 | 10.28±0.32 | 56.77±0.40 | 10.19±0.36 | 56.61±0.30 | 10.21±0.32 |
| FML [arXiv20] | 46.61±0.61 | 15.18±0.34 | 46.16±1.02 | 15.78±0.75 | 46.04±1.16 | 15.85±0.83 |
| FedGen [ICML21] | 54.27±0.15 | 10.29±0.26 | 54.38±0.02 | 10.33±0.31 | 54.50±0.18 | 10.45±0.46 |
| FedKD [NC22] | 54.86±0.43 | 10.43±0.23 | 54.82±0.37 | 10.36±0.33 | 54.55±0.13 | 10.45±0.21 |
| FedAPEN [KDD23] | 60.30±0.33 | 9.40±0.30 | 60.35±0.31 | 9.45±0.31 | 60.40±0.31 | 9.40±0.30 |
| FedTGP [AAAI24] | 53.39±0.53 | 9.73±0.61 | 53.40±0.52 | 10.62±1.17 | 52.85±1.16 | 10.28±0.78 |
| FedMRL [NIPS24] | 52.80±0.63 | 12.81±1.17 | 52.61±0.77 | 12.85±1.20 | 53.11±0.63 | 12.95±1.27 |
| **PHP-FL (Ours)** | **66.85±0.36** | **8.07±0.25** | **66.94±0.39** | **8.07±0.24** | **66.78±0.36** | **8.09±0.27** |
| | ↑ **6.55** | ↓ **1.38** | ↑ **6.59** | ↓ **1.29** | ↑ **6.38** | ↓ **1.31** |

## 5.2 Comparison to State-of-the-Arts Methods

We evaluate PHP-FL against several state-of-the-art methods in Tables 1 and 2. The experiments are conducted under three distinct client participation patterns: *uniform*, *normal*, and *linear*, representing diverse real-world scenarios. Across all settings, PHP-FL consistently demonstrates superior performance. It achieves the highest average accuracy (highest AM values) while simultaneously exhibiting the best fairness (lowest FM values). Compared to the strongest baseline FedAPEN, PHP-FL achieves notable improvements in average performance, boosting AM by up to $0.87\%$ on Fashion-MNIST and a substantial $6.51\%$ on CIFAR-10. At the same time, it enhances fairness by reducing FM by up to $2.48\%$ and $1.33\%$ on the respective datasets, demonstrating the robustness and effectiveness of PHP-FL in addressing model heterogeneity and unfairness caused by inconsistent client participation. Appendix C.1 further shows its faster convergence and superior performance through accuracy and standard deviation curves.

## 5.3 Ablation Study

In Table 3, we present an ablation study to evaluate the contribution of the DEAL and ISPU modules in PHP-FL under the *normal* participation pattern. When both modules are disabled, the performance significantly degrades, especially on CIFAR-10, where the accuracy drops to 59.88%.

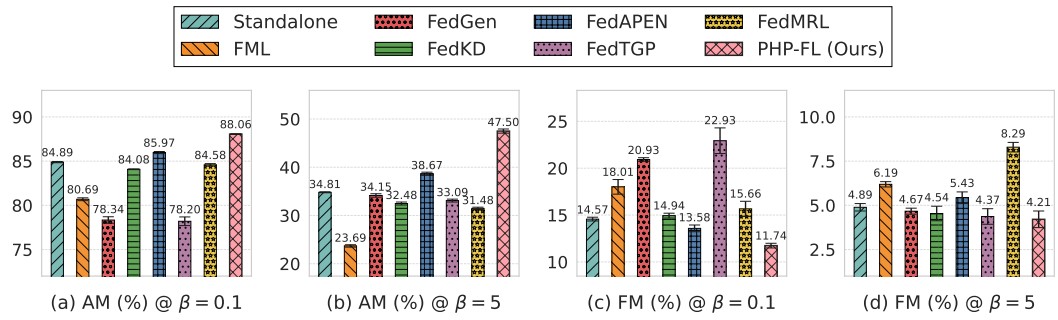

Figure 3: Comparison results on CIFAR-10 under varying degrees of data distribution heterogeneity across clients. All other settings follow their default configurations.

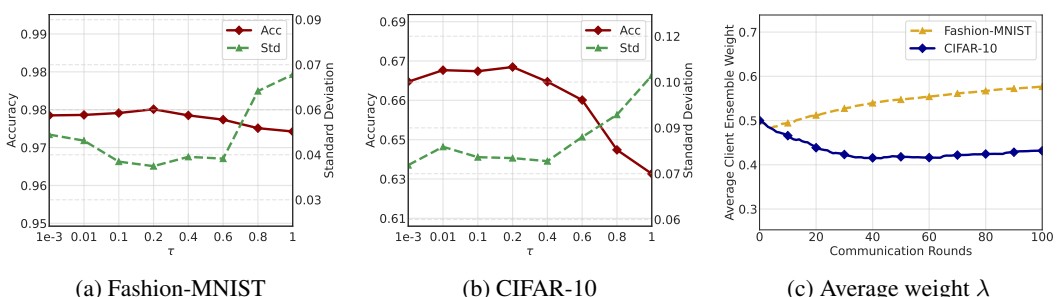

(a) Fashion-MNIST      (b) CIFAR-10      (c) Average weight $\lambda$

Figure 4: Left two: effect of $\tau$ on performance. Right: average weight $\lambda$ of clients in each round.

Introducing the ISPU module alone brings a modest improvement, highlighting its effectiveness in mitigating update conflicts from inconsistent client participation.

Besides, enabling only the DEAL module yields a more pronounced performance gains. This is achieved by effectively addressing system heterogeneity through data-free knowledge alignment and ensemble learning. Notably, enabling both DEAL and ISPU achieves the best performance on both datasets, with 97.59% accuracy on Fashion-MNIST and 66.94% on CIFAR-10, demonstrating their complementarity and the necessity of their joint design.

Table 3: **Ablation study** of key modules of PHP-FL under the *normal* pattern.

| DEAL | ISPU | Fashion-MNIST | | CIFAR-10 | |
|---|---|---|---|---|---|
| | | AM (%) | FM (%) | AM (%) | FM (%) |
| ✗ | ✗ | 92.86 | 9.10 | 59.88 | 11.24 |
| ✗ | ✓ | 96.72 | 5.07 | 62.05 | 9.47 |
| ✓ | ✗ | 96.98 | 5.73 | 63.08 | 8.16 |
| ✓ | ✓ | **97.59** | **4.24** | **66.94** | **8.07** |

## 5.4 Case Studies

**Robustness to Non-IIDness.** To evaluate PHP-FL's robustness under varying data heterogeneity, we conducted additional experiments on CIFAR-10 using the Dirichlet distribution with $\beta = 0.1$ (high heterogeneity) and $\beta = 5$ (low heterogeneity). As shown in Figure 3, PHP-FL consistently outperforms all baselines across both settings. Under high heterogeneity, PHP-FL surpasses the best-performing baseline by 2.09% in accuracy (AM) and reduces the fairness metric (FM) by 1.84%. This advantage becomes even more pronounced under low heterogeneity, where PHP-FL achieves a remarkable 8.83% accuracy gain over the next best method while maintaining the best fairness performance. These results clearly show that PHP-FL is not only robust to different levels of data heterogeneity but consistently achieves state-of-the-art performance in both accuracy and fairness.

**Effect of the Pruning Threshold $\tau$ on Performance.** To investigate the effect of the hyperparameter $\tau$, we conduct experiments on Fashion-MNIST and CIFAR-10 under the *normal* pattern. As shown in Figure 4a and 4b, on the CIFAR-10 dataset, the accuracy first increases and then decreases, while the standard deviation initially decreases and then increases, with both metrics achieving their best values when $\tau$ is set to 0.2. For the Fashion-MNIST dataset, the performance remains relatively

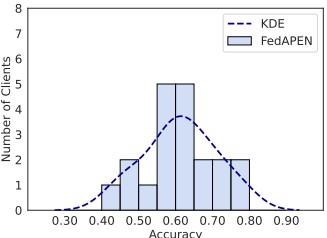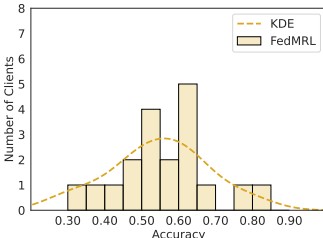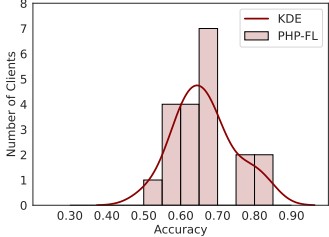

Figure 5: The client accuracy distribution at the best achieved AM (best mean accuracy) on CIFAR-10 dataset under the *uniform* participation pattern for PHP-FL and two other baseline methods.

stable across different $\tau$, and similarly, the best results are also observed at $\tau = 0.2$. Therefore, we choose $\tau = 0.2$ as the default configuration for all experiments.

**Effect of Adaptive Ensemble Weights.** We analyze the behavior of the adaptive ensemble weight $\lambda$ by tracking its average value across clients throughout training under the *uniform* pattern. As depicted in Figure 4c, the dynamics of $\lambda$ differ significantly between datasets. For CIFAR-10, the average $\lambda$ is initialized as $0.5$ but quickly decreases and stabilizes around $0.43$, indicating a consistent preference for the global model $g$ within the ensemble on this more complex dataset. In contrast, on Fashion-MNIST, the average $\lambda$ steadily increases from $0.5$ to approximately $0.58$ by the end of training, signifying a growing reliance on the specialized local models $w$. This demonstrates that the adaptive mechanism effectively captures dataset-specific characteristics, dynamically adjusting the ensemble balance between local and global models to leverage their respective strengths during the learning process.

**Visualization of Client Accuracy Distribution.** To visualize the client accuracy distribution under the *normal* participation pattern, we plot histograms and Kernel Density Estimation (KDE) [66] curves for different methods on CIFAR-10 dataset. As shown in Figure 5, PHP-FL achieves a more concentrated accuracy distribution compared to FedAPEN and FedMRL, with clients generally attaining higher accuracy. Moreover, the differences in client performance are significantly reduced under PHP-FL, highlighting its superiority in both enhancing average performance and promoting fairness across clients.

## 6  Conclusion

In this paper, we propose PHP-FL, a novel model-heterogeneous federated learning method addressing the critical challenge of enhancing both accuracy and fairness under inconsistent client participation probabilities. PHP-FL achieves this through two integrated modules: (1) the DEAL module, which harmonizes heterogeneous models via data-free knowledge alignment; and (2) the ISPU module, which selectively updates task-relevant parameters to mitigate update conflicts. Evaluated across diverse participation patterns, PHP-FL demonstrates state-of-the-art performance for both accuracy and fairness. Ablation study further validates the effectiveness of each module. Our research narrows the divide between idealized uniform participation scenarios and practical heterogeneous FL systems, providing a lightweight yet robust solution suitable for real-world implementation.

**Limitations.** Despite the promising results, PHP-FL has two main limitations:

First, compared to approaches that exchange only lightweight information (*e.g.*, logits, prototypes [29, 12], or partial model parameters [10, 11]), our method introduces non-negligible computation and communication overheads.[6] Although employing a smaller auxiliary model can alleviate this burden, the additional costs from ensemble training and selective parameter updates still persist.

Second, our evaluation is conducted on a constrained set of model heterogeneity types, datasets, and client participation patterns. Although PHP-FL demonstrates robust performance within these scenarios, its generalizability should be further verified on more diverse and large-scale benchmarks.

---

[6]For an analysis of the communication and computation costs, please refer to Appendix C.7.

## Acknowledgments and Disclosure of Funding

This work was supported by the National Key Research and Development Program of China under Grant No.2024YFE0204500, and in part by the National Natural Science Foundation of China under Grant No.92267104.

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

# APPENDIX

## A  Pseudo codes of PHP-FL

The algorithm is outlined in Algorithm 1. Please refer to Section 4.1 for more details.

---

**Algorithm 1:** PHP-FL

---

**Input:** Auxiliary small models $\{g_i^0\}_{i=1}^K$, Local models $\{w_i^0\}_{i=1}^K$, Total number of rounds $T$,
       Dataset partitions $\{D_i\}_{i=1}^K$.
**Output:** Optimized local models $\{w_i^T\}_{i=1}^K$ and auxiliary small models $\{g_i^T\}_{i=1}^K$.
Initialize historical mask $\{M_i^h\}_{i=1}^K \leftarrow \mathbf{1}^{|g|}$ and global model $\mathcal{G}^0$;
**for** *round $t = 1$ to $T$* **do**
  | **Server Side:**
  |   Collect the IDs of active clients $\mathcal{A}_t \subseteq \{1, ..., K\}$;
  |   Broadcast the pruned auxiliary small model $\mathcal{G}^{t-1} \odot M_i^h$ to client $i \in \mathcal{A}_t$;
  |   $\{g_i^t, M_i^t\}_{i\in\mathcal{A}_t} \leftarrow$ **Client Update**;
  |   Update the historical mask $M_i^h \in M_{\text{hist}}$ for clients in $\mathcal{A}_t$: $M_i^h \leftarrow M_i^t$.
  |   Aggregate auxiliary small models: $\mathcal{G}^t = \frac{1}{|\mathcal{A}_t|} \sum_{i\in\mathcal{A}_t} g_i^t$;
  | **Client Update:**
  | **for** *each client $i \in \mathcal{A}_t$ in parallel* **do**
  |  |   Download $\mathcal{G}^{t-1} \odot M_i^h$ from server;
  |  |   $\hat{g}_i^{t-1} \leftarrow$ initialize local auxiliary model by Eq. 11;
  |  |   Randomly divides the training set $D_i$ into $D_i^a$ and $D_i^s$;
  |  |   **for** *batch $(x, y) \in D_i^a$* **do**
  |  |  |   $\lambda_i^t \leftarrow$ update ensemble weight by Eq. 5;
  |  |   **end**
  |  |   **for** *batch $(x, y) \in D_i^s$* **do**
  |  |  |   $\mathcal{L}_w \leftarrow$ Calculate local training loss by Eq. 6;
  |  |  |   $\mathcal{L}_g \leftarrow$ Calculate the symmetrical loss similar to Eq. 6;
  |  |  |   $w_i^t, g_i^t \leftarrow$ update the local model and local auxiliary model by Eq. 7;
  |  |   **end**
  |  |   $\alpha_i^t \leftarrow$ calculate update ratio by Eq. 9;
  |  |   $M_i^t \leftarrow$ obtain binary mask by Eq. 10 ;
  |  |   Upload $g_i^t$ and the binary mask $M_i^t$ to server;
  | **end**
**end**

---

## B  Additional Experimental Details

### B.1  Hyperparameter Settings

We provide a detailed summary of the hyperparameter configurations used in our experiments in Table 4. These settings are carefully selected to ensure fair comparison across different baselines. [7] For the proposed PHP-FL, the standardized feature dimension $d$ is set to 512. The adaptability set $D_i^a$ consists of 10% randomly sampled data from the training set $D_i$. The ensemble weight $\lambda_i^t$ is trained for 10 epochs in each round. Additionally, following the hyperparameter search detailed in Section 5.4 and C.6, the pruning threshold $\tau$ and the sharpness factor $\delta$ are set to 0.2 and 5, respectively. Unless otherwise specified, all experiments follow the same training setting.

---

[7]Note that these parameter names in different methods are consistent with the original references and are independent of the notation used in our work.

Table 4: Hyperparameter settings used in our experiments.

| Type | Hyperparameters | Value |
|------|-----------------|-------|
| FL training settings | Local learning rate $\eta$ | 0.001 |
| | Batch size | 64 |
| | Local epochs per round $E$ | 5 |
| | Total rounds $T$ | 100 |
| | Number of clients $K$ | 20 |
| Framework-specific | $\alpha$ in FML | 0.5 |
| | $\beta$ in FML | 0.5 |
| | Server learning epochs in FedGen | 100 |
| | Server learning rate in FedGen | 0.1 |
| | $d_\eta$ in FedGen | 32 |
| | $d_h$ in FedGen | 512 |
| | $T_{start}$ in FedKD | 0.95 |
| | $T_{end}$ in FedKD | 0.98 |
| | $\eta_s$ in FedKD | 0.001 |
| | Adaptation set ratio in FedAPEN | 10% |
| | Server learning epochs in FedTGP | 100 |
| | $\tau$ in FedTGP | 100 |
| | $\lambda$ in FedTGP | 0.1 |
| | $d_1$ in FedMRL | 128 |
| | $\lambda$ in Ditto | 0.1 |
| | $\alpha$ for CIFAR-10 in FedFV | 0.1 |
| | $\alpha$ for Fashion-MNIST in FedFV | 0 |
| | $\tau$ for CIFAR-10 in FedFV | 10 |
| | $\tau$ for Fashion-MNIST in FedFV | 0 |
| | $\beta$ in FedHEAL | 0.4 |
| | $\tau$ in FedHEAL | 0.4 |
| | $K$ in FedAU | 1 |

## B.2 Visualization of Data Distributions

To intuitively illustrate the data heterogeneity across clients in our federated learning setting, we plot scatter diagrams based on the CIFAR-10 and Fashion-MNIST datasets in Figure 6. Specifically, we simulate heterogeneous data distributions by allocating the proportion of class $j$ to each client $k$ according to a Dirichlet distribution ($p_{j,k} \sim \text{Dir}(\beta)$), where a smaller $\beta$ indicates more extreme data heterogeneity across clients. In our experiments, we set $\beta = 0.1$ for Fashion-MNIST and $\beta = 0.5$ for CIFAR-10, respectively.

## B.3 Model Architectures Used in Experiments

we utilize four widely recognized Convolutional Neural Network (CNN) architectures with varying designs and complexities. We report the corresponding parameter counts of each model in Table 5. These serve as the local models for clients in our heterogeneous federated learning setup:

- **GoogLeNet** [60]: Introduced the inception module, which performs convolutions with multiple filter sizes in parallel within the same block. It was designed for computational efficiency and won the ILSVRC 2014 challenge.

- **DenseNet-121** [61]: Characterized by its dense connectivity pattern, where each layer receives feature maps from all preceding layers within a dense block. This encourages feature reuse, strengthens gradient flow, and improves parameter efficiency. The '121' denotes the number of layers with weights.

- **EfficientNet-B1** [62]: Developed using neural architecture search and a compound scaling method that uniformly scales network width, depth, and resolution. It aims to balance model accuracy with computational efficiency (FLOPS and parameters). B1 is a specific scaled version providing a good trade-off.

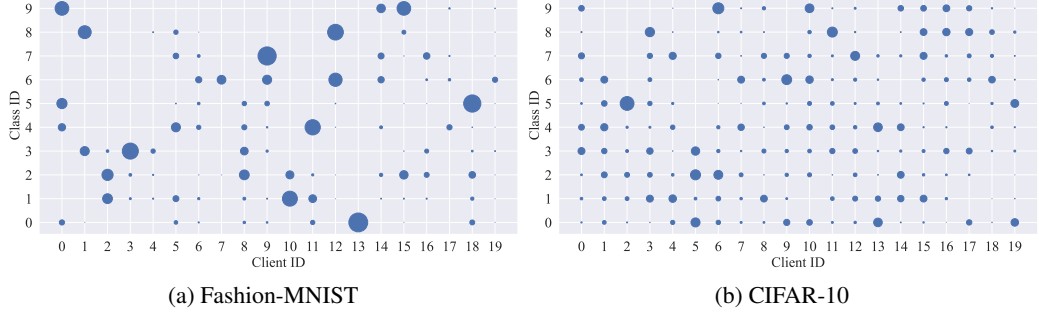

(a) Fashion-MNIST                           (b) CIFAR-10

Figure 6: The data distribution of 20 clients in our experiments.

- **ResNet-18** [63]: Employs residual connections (skip connections) that allow gradients to bypass layers, enabling the training of much deeper networks by mitigating the vanishing gradient problem. ResNet-18 is a relatively shallow variant with 18 layers containing weights.

For a fair comparison, we adopt GoogLeNet [60] as the auxiliary model architecture for all methods that utilize an auxiliary model, including our proposed method PHP-FL, since it has the smallest number of parameters among the four candidate models.

Table 5: Parameter counts of the evaluated models. "M" is short for million.

| Model | Parameter counts |
|---|---|
| GoogleNet [60] | 5.61M |
| DenseNet-121 [61] | 6.96M |
| EfficientNet-B1 [62] | 6.52M |
| ResNet-18 [63] | 11.18M |

## C  Additional Experimental Results

### C.1  Comparison of Training Curves for Accuracy and Standard Deviation

To evaluate the convergence behavior and training stability of PHP-FL, we compare the training curves in terms of average accuracy and standard deviation across training rounds on Fashion-MNIST and CIFAR-10 under the *uniform* pattern. As shown in Figure 7 and 8, the results on both datasets demonstrate that the proposed method PHP-FL significantly accelerates convergence compared to baselines, while maintaining a low and stable standard deviation throughout training, which effectively enhances performance and fairness across clients.

### C.2  Comparison to More State-of-the-Arts Methods in the Homogeneous Setting

To further validate the generality and effectiveness of PHP-FL, we conduct additional experiments in the homogeneous setting and report the results in Table 6. Specifically, all clients adopt the identical GoogLeNet architectures [60]. Additional compared methods include FedFV [41] and Fed-HEAL [28], which are designed to improve client fairness (**Fair-FL**), as well as FedAWE [20] and FedAU [45], which address client unavailability (**CU-FL**). As shown in Table 6, PHP-FL achieves the highest average accuracy (AM) across all patterns, while also maintaining a highly competitive performance fairness (FM). Although FedHEAL exhibits slightly better fairness, its average accuracy is significantly inferior compared to PHP-FL. This highlights that PHP-FL not only delivers the highest average performance but also achieves fairness that is highly competitive with the best fair-FL methods, striking an exceptional balance that surpasses existing methods in overall effectiveness even in the homogeneous setting. These results underscore the robustness and practicality of PHP-FL even in homogeneous FL scenarios, reaffirming its overall superiority.

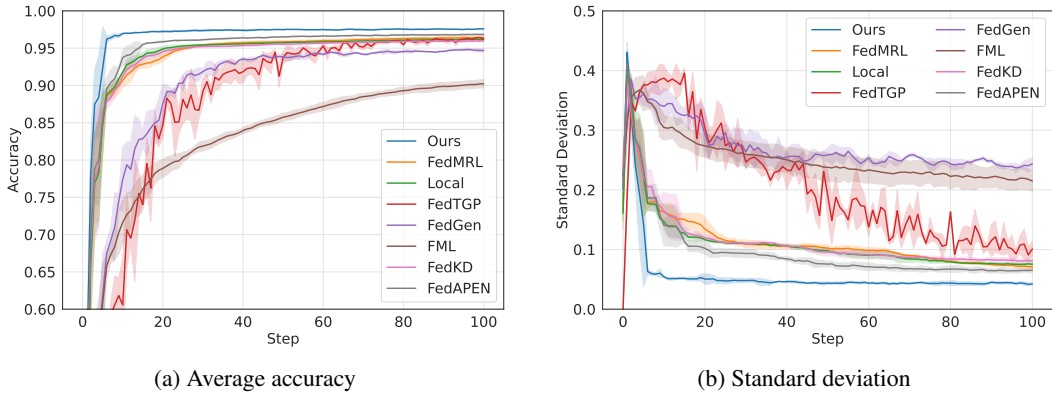

(a) Average accuracy           (b) Standard deviation

Figure 7: Comparison of training curves on Fashion-MNIST.

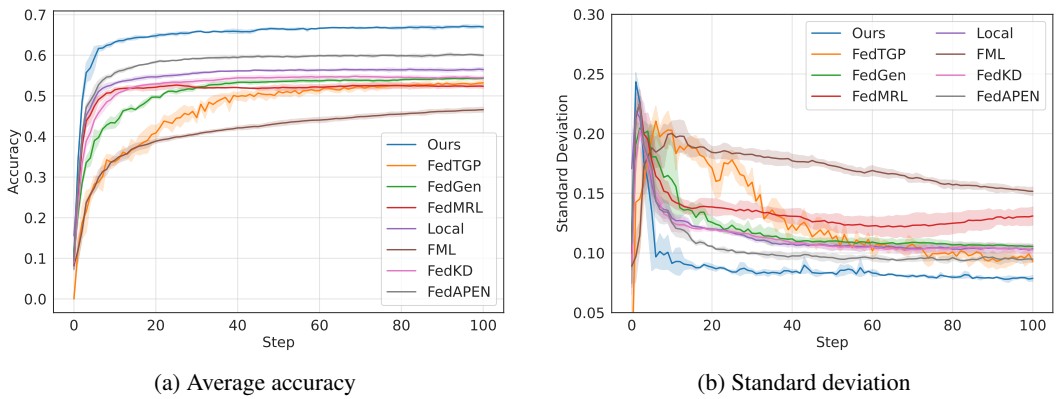

(a) Average accuracy           (b) Standard deviation

Figure 8: Comparison of training curves on CIFAR-10.

Table 6: Comparison with the state-of-the-art methods on CIFAR-10 in the homogeneous setting. Best in **bold** and second with underline.

| Type | Methods | *Uniform* $[a = 0.5]$ | | *Normal* $[\mu = 0.5, \sigma = 0.2]$ | | *Linear* $\left[a = 0.05, d = \frac{K-2}{K(K-1)}\right]$ | |
|------|---------|---------|---------|---------|---------|---------|---------|
| | | AM (%) ↑ | FM (%) ↓ | AM (%) ↑ | FM (%) ↓ | AM (%) ↑ | FM (%) ↓ |
| **MH-FL** | FML [arXiv20] | 41.31±0.46 | 13.28±0.31 | 41.00±0.48 | 13.46±0.35 | 39.21±2.80 | 13.37±0.30 |
| | FedGen [ICML21] | 61.79±0.13 | 8.80±0.46 | 61.73±0.15 | 8.95±0.25 | 61.60±0.29 | 8.99±0.20 |
| | FedKD [NC22] | 61.26±0.08 | 10.03±0.10 | 60.96±0.38 | 9.93±0.11 | 60.73±0.70 | 10.01±0.09 |
| | FedAPEN [KDD23] | 66.49±0.30 | 8.14±0.16 | 66.54±0.31 | 8.26±0.18 | 66.60±0.35 | 8.25±0.17 |
| | FedTGP [AAAI24] | 60.12±0.48 | 9.97±0.46 | 60.15±0.46 | 10.20±0.31 | 59.88±0.69 | 10.14±0.31 |
| | FedMRL [NIPS24] | 62.86±0.62 | 8.60±0.16 | 62.86±0.62 | 8.95±0.58 | 62.58±0.54 | 8.80±0.38 |
| **Fair-FL** | FedFV [IJCAI21] | 59.75±0.87 | 11.03±1.63 | 60.05±1.31 | 10.54±2.02 | 58.32±1.45 | 11.08±1.60 |
| | FedHEAL [CVPR24] | 59.43±0.83 | **7.03±0.76** | 59.36±0.75 | **7.93±1.14** | 59.15±0.57 | **7.94±1.15** |
| **CU-FL** | FedAWE [NIPS24] | 62.75±0.09 | 9.39±0.30 | 62.68±0.13 | 9.31±0.25 | 62.54±0.31 | 9.36±0.28 |
| | FedAU [ICLR24] | 62.79±0.20 | 9.17±0.19 | 62.72±0.25 | 9.21±0.29 | 62.59±0.47 | 9.35±0.15 |
| **All** | **PHP-FL (Ours)** | **67.99±0.16** | 8.05±0.19 | **67.88±0.32** | 8.08±0.16 | **67.84±0.13** | 8.12±0.12 |

## C.3 Performance with Varying Numbers of Clients

In this Section, we compare the performance of different methods with varying numbers of clients on CIFAR-10. Specifically, the *uniform* pattern involves 10 clients with full participation in every round; the *normal* pattern consists of 50 clients whose participation probabilities are sampled from a normal distribution ($\mu = 0.2$, $\sigma = 0.2$), resulting in an average of 10 clients per round; and the *linear* pattern also includes 50 clients, with participation probabilities increasing linearly from 0.02 by a step of $\frac{0.36}{K-1}$, yielding the same average of 10 clients per round. As shown in Table 7, PHP-FL consistently achieves the best accuracy (AM) and performance fairness (FM) across all three patterns. Notably, compared to the strongest baseline FedAPEN, our method improves AM by 6.62%, 6.47%, and 6.12% under the *uniform*, *normal*, and *linear* settings, respectively, while also reducing FM, demonstrating superior robustness and fairness with varying numbers of clients.

Table 7: Comparison with the state-of-the-art methods with varying numbers of clients on CIFAR-10 in the heterogeneous setting. Best in **bold** and second with underline.

| Methods | Uniform [$a = 1.0$] | | Normal [$\mu = 0.2, \sigma = 0.2$] | | Linear $\left[ a = 0.02, d = \frac{0.36}{K-1} \right]$ | |
|---|---|---|---|---|---|---|
| | AM (%) ↑ | FM (%) ↓ | AM (%) ↑ | FM (%) ↓ | AM (%) ↑ | FM (%) ↓ |
| Standalone | 58.89±0.20 | 10.53±0.24 | 50.73±1.29 | 14.04±1.40 | 51.81±0.30 | 12.81±0.47 |
| FML [arXiv20] | 46.61±0.61 | 15.18±0.34 | 36.59±1.17 | 17.35±1.22 | 35.26±0.22 | 17.10±0.16 |
| FedGen [ICML21] | 57.82±0.06 | 9.45±0.29 | 43.70±1.03 | 15.10±1.26 | 43.35±1.08 | 14.92±0.53 |
| FedKD [NC22] | 57.46±0.45 | 11.26±0.33 | 49.27±0.93 | 13.98±0.63 | 49.07±0.62 | 13.99±0.71 |
| FedAPEN [KDD23] | 63.90±0.10 | 9.05±0.51 | 52.99±1.23 | 13.68±1.77 | 53.76±0.39 | 12.74±0.27 |
| FedTGP [AAAI24] | 58.45±0.69 | 9.88±0.37 | 40.15±0.32 | 14.74±0.69 | 38.18±1.04 | 16.08±1.48 |
| FedMRL [NIPS24] | 58.33±0.20 | 14.23±0.63 | 49.32±1.34 | 13.64±1.01 | 49.26±0.74 | 13.26±0.27 |
| **PHP-FL (Ours)** | **70.52±0.03** | **7.97±0.49** | **59.46±0.83** | **12.69±1.41** | **59.88±0.26** | **11.22±0.37** |

Table 8: Comparison results on CIFAR-10 dataset under the *Markovian* participation pattern. All other settings follow their default configurations.

| Method | Standalone | FML | FedGen | FedKD | FedAPEN | FedTGP | FedMRL | PHP-FL (Ours) |
|---|---|---|---|---|---|---|---|---|
| AM (%) ↑ | 52.77±1.44 | 42.73±0.96 | 51.93±0.84 | 51.31±0.80 | 57.84±0.53 | 50.55±1.79 | 49.02±1.74 | **64.04±0.73** |
| FM (%) ↓ | 16.10±2.50 | 19.29±2.04 | 14.91±0.38 | 16.82±1.45 | 13.99±3.34 | 12.14±1.66 | 15.86±0.08 | **11.69±1.35** |

## C.4 Performance under the *Markovian* Participation Pattern

To better reflect dynamic client availability in real-word scenarios, we have conducted additional experiments under the *Markovian* participation pattern following FedAU [45]. In this pattern, each client follows a two-state Markov chain to determine its participation status in each training round, where the two states correspond to "participating" and "not participating" in the current training round. This modeling introduces temporal correlation in client participation behavior while maintaining sufficient randomness, offering a more realistic simulation compared to independently sampled participation patterns.

For the parameter settings of the *Markovian* pattern experiment, we constrain the maximum transition probability from the non-participating state to the participating state to 0.05, thereby avoiding excessively frequent state changes and ensuring realistic participation dynamics. The initial state of each client's Markov chain is randomly sampled according to the stationary probability, which is set to 0.5 to ensure that approximately half of the clients participate in each round on average. The transition probabilities are carefully calibrated to maintain this stationary distribution throughout the training process, ensuring system stability while introducing participation heterogeneity across clients. As shown in Table 8, the experimental results under the *Markovian* participation pattern further validate PHP-FLs robustness, consistently achieving superior accuracy and fairness despite the increased dynamic and unpredictable client availability.

## C.5 Effectiveness of Adaptive Selective Updates in the ISPU Module

To further validate the design rationality of the adaptive selective parameter update mechanism in our ISPU module, we conduct an ablation study comparing different update strategies for the local auxiliary model $g$, focusing on the update ratio $\alpha$. Specifically, we evaluate the following variants:

- **Fixed update.** Updates a fixed ratio of the most important parameters, with $\alpha$ set to a constant and we adopt $\alpha = 0.5$.
- **Full Update.** Updates all parameters without selection, which is equivalent to $\alpha = 1$.
- **Random Update.** All parameters are stochastically updated with probability $\alpha$, where $\alpha$ is dynamically determined by our proposed method using Eq. 9.
- **Adaptive update (Ours).** Dynamically adjusts $\alpha$ using Eq. 9 based on client participation history and parameter importance.

As shown in Table 9, our adaptive update mechanism consistently achieves the best accuracy and performance fairness. Full update shows the poorest performance, and while fixed Update offers some stability, it is surpassed by the adaptive methods. Random update achieves competitive accuracy but inferior fairness compared to our method. The findings highlight the superiority of our adaptive selective update mechanism within the ISPU module.

Table 9: Effectiveness of Adaptive Selective Updates in the ISPU module. Results are evaluated under the *linear* participation pattern on Fashion-MNIST and CIFAR-10. Best in **bold**.

| Different variants | Fashion-MNIST | | CIFAR-10 | |
|---|---|---|---|---|
| | AM (%) ↑ | FM (%) ↓ | AM (%) ↑ | FM (%) ↓ |
| **Fixed Update** ($\alpha = 0.5$) | 97.43 | 4.41 | 64.11 | 9.14 |
| **Full Update** ($\alpha = 1.0$) | 96.82 | 5.73 | 62.81 | 8.65 |
| **Random Update** (dynamic $\alpha$) | 97.28 | 6.23 | 65.97 | 8.97 |
| **Adaptive Update** (**Ours** with dynamic $\alpha$) | **97.58** | **4.29** | **66.78** | **8.09** |

## C.6 Effect of the Sharpness Factor on Performance

In our proposed PHP-FL, the hyperparameter $\delta$ in Eq. 9 controls the sharpness of the mapping from local participation frequency to the update ratio $\alpha$ in the ISPU module. A larger $\delta$ causes $\alpha$ to approach 1 for frequently participating clients and 0 for infrequent ones, whereas a smaller $\delta$ smooths the adjustment, pushing $\alpha$ toward 0.5. As shown in Table 10, performance is relatively stable across a range of $\delta$ values, but we observe that $\delta = 5$ consistently achieves near-optimal results in both accuracy and fairness across Fashion-MNIST and CIFAR-10. Therefore, we set $\delta = 5$ in our experimental configuration.

Table 10: Effect of the hyperparameter $\delta$ on performance. Best in **bold** and second with underline.

| Dataset | Metric | $\delta = 0.1$ | $\delta = 0.5$ | $\delta = 1$ | $\delta = 5$ | $\delta = 10$ | $\delta = 50$ | $\delta = 100$ |
|---|---|---|---|---|---|---|---|---|
| Fashion-MNIST | AM (%) ↑ | 97.54 | **97.62** | 97.59 | 97.61 | 97.59 | 97.52 | 97.51 |
| | FM (%) ↓ | 4.25 | 4.13 | 4.39 | **4.12** | 4.60 | 4.35 | 4.43 |
| CIFAR-10 | AM (%) ↑ | 67.24 | **67.28** | 67.06 | 67.27 | 66.36 | 67.17 | 66.82 |
| | FM (%) ↓ | 8.44 | 8.10 | 8.12 | **8.01** | 8.51 | 8.49 | 8.55 |

## C.7 Cost and Efficiency Analysis

**Communication Cost.** We compare the per-round communication cost with baselines in terms of the number of parameters transmitted between 20 clients and the server on CIFAR-10. As shown in Table 11, among the evaluated baselines, methods such as FedGen and FedTGP demonstrate significantly lower communication costs due to their use of partial model sharing and lightweight

prototypes. Unfortunately, methods relying on auxiliary model transmission (*e.g.*, FML, FedKD, FedAPEN, and FedMRL) exhibit communication cost exceeding 200M parameters per round. In contrast, PHP-FL requires only 112.52M parameters per round, which is nearly half the cost of other auxiliary model-based approaches such as FedAPEN and FedMRL. This reduction is primarily due to clients downloading only the pruned global auxiliary model parameters instead of the full model.

**Computation Cost.**  We also report the total computation cost per round across all clients in terms of FLOPs (floating-point operations),[8] as summarized in Table 11. Following [12], other operations such as data preprocessing are not included in the FLOPs calculation. To ensure a fair comparison, this experiment involves 20 clients with full participation in each round. All other configurations remain aligned with the main experiments.

According to Table 11, PHP-FL incurs the per-round computation cost at 813.71GFLOPs. This increase is marginal when compared with other auxiliary model-based methods such as FML (753.51G), FedKD (753.51G), FedAPEN (771.01G), and FedMRL (757.34G). Compared to Fed-Gen (391.38G) and FedTGP (387.99G), the increased computation cost primarily arises from the training of additional auxiliary models. Besides, the slightly higher cost of the proposed PHP-FL is mainly attributed to the extra training required for ensemble weights. While FedTGP achieves the lowest computation cost, its accuracy significantly lags. Despite this slight increase in cost compared with other auxiliary model-based methods, PHP-FL achieves significantly the best results in both accuracy and performance fairness, as shown in the main experiments. The results underscore a compelling trade-off, with our method delivering notable performance gains at the cost of only a slight increase in computation.

Table 11: Comparison of communication and computation costs per round on CIFAR-10, where communication cost is measured by the number of parameters transmitted between 20 clients and the server, and computation cost is evaluated as the total number of FLOPs executed across all 20 clients. 'M' and 'G' denote million and giga, respectively. For FedKD, the SVD computation cost is excluded from this analysis.

| Method | Communication Cost | Computation Cost |
|---|---|---|
| FML [arXiv20] | 224.20M | 753.51G |
| FedGen [ICML21] | 5.92M | 391.38G |
| FedKD [NC22] | 200.26M | 753.51G |
| FedAPEN [KDD23] | 224.20M | 771.01G |
| FedTGP [AAAI24] | 0.192M | 387.99G |
| FedMRL [NeurIPS24] | 224.05M | 757.34G |
| PHP-FL (Ours) | 112.52M | 813.71G |

**Efficiency Analysis.**  In addition, we conducted further experiments to measure the total costs (communication rounds, computation cost, and communication cost) required to reach 50% accuracy compared to baselines. As shown in Figure 9, while PHP-FL's per-round communication cost is higher than non auxiliary model-based methods (*e.g.*, FedGen and FedTGP) due to the transmission of auxiliary model, it reaches target accuracy in just 2 rounds, whereas all baselines require at least 5 rounds. Consequently, the total communication and computation costs are substantially lower. Furthermore, the per-round costs can be readily optimized in practice by employing smaller auxiliary models or mask quantization (*e.g.*, bitwidth reduction). Thus, PHP-FL offers a far more efficient path to convergence in practical deployments.

# D  Discussion

**Communication Cost.**  In the auxiliary model-based methods, active clients often need to upload $L = |g|$ parameters of the global auxiliary model in each communication round, which are typically represented in full-precision floating-point format. Our method similarly requires uploading

---

[8]We calculate FLOPs using the thop library: https://github.com/Lyken17/pytorch-OpCounter.

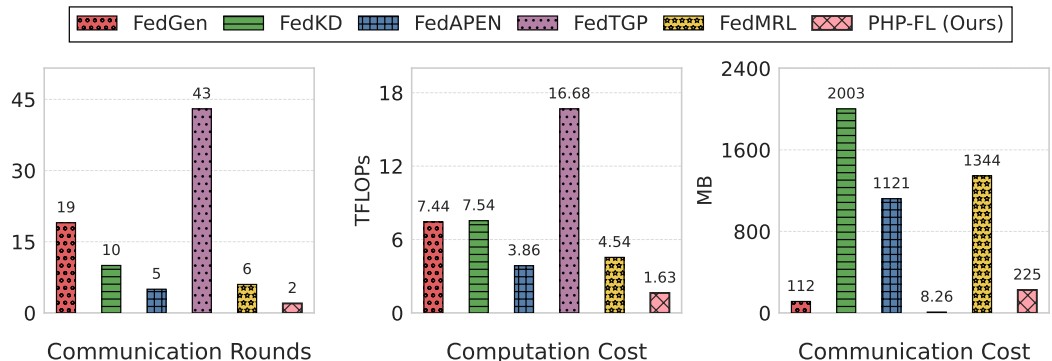

Figure 9: From left to right: Communication rounds, total number of transmitted parameters, and computation FLOPs required to achieve the 50% accuracy on CIFAR-10 under the *uniform* pattern.

the global auxiliary model but with an additional binary mask matrix $M$ of the same dimension $L$. Fortunately, since each element in $M$ is a binary value (0 or 1), it only incues **1 bit per parameter**, rendering the added communication overhead negligible compared to the transmission of full-precision parameters. Moreover, when downloading the global auxiliary model, clients only need to receive $\alpha \cdot L$ ($\alpha \in (0, 1)$) parameters, where the pruning threshold $\tau$ in Eq 9 can be adjusted as needed. This flexibility allows us to further reduce communication costs dynamically.

**Participation Patterns.** We clearly state that we make no assumptions about the distribution of client participation patterns and allow them to be arbitrary throughout the training process. Moreover, similar to [45], we do not require any prior knowledge of the client sampling process for the proposed method PHP-FL.

**Privacy.** Similar to FedAvg [1], PHP-FL does not require sharing raw data or client-specific heterogeneous model parameters. Instead, only the lightweight, homogeneous auxiliary models and the mask matrix related to model parameters are uploaded to the server. This design ensures that sensitive local model structures and data remain on the client side, making PHP-FL suitable for privacy-critical applications. Therefore, PHP-FL is compatible with standard privacy-preserving mechanisms such as secure aggregation [67]. Differentially private variants of FedAvg [68] can be seamlessly integrated similarly.

