# OpenReview forum: "A Fair Federated Learning Method for Handling Client Participation Probability Inconsistencies in Heterogeneous Environments"
_NeurIPS.cc/2025/Conference — NeurIPS 2025 poster_

### Official Review · Reviewer_VyUg · 2025-06-06

**Clarity:** 3
**Significance:** 2
**Originality:** 3
**Rating:** 4
**Confidence:** 3

**Summary:**

This paper addresses the challenge of inconsistent client participation in Federated Learning. The proposed approach consists of two key modules: the DEAL module, which employs a homogeneous auxiliary model to align heterogeneous local models and facilitate ensemble learning for improved local accuracy, and the ISPU module, which adaptively updates task-relevant parameters using an importance-based masking mechanism. The authors support their method with experimental results.

**Questions:**

1: In the introduction (line 27), the authors mention that it is impractical to assume all clients share an identical global model. However, the DEAL module still relies on a small homogeneous auxiliary model. Can this assumption be relaxed? If not, what are the key challenges in doing so?

2: In Figure 3c, the values of $\lambda$ for both datasets appear not to converge. Could the authors clarify whether convergence is expected?

3: The paper lacks theoretical analysis.

Minor:

1: According to Definition 3, should the linear pattern plot in Figure 1c be sorted in ascending order to better reflect the defined metric?

2: In line 127, the sentence is imprecise. The variable $p_i$ implicitly depends on other clients since it is defined as a function of $i$ (the client index), which may introduce cross-client dependencies.

**Ethical Concerns:**

["NO or VERY MINOR ethics concerns only"]

**Final Justification:**

The authors have thoroughly addressed my concerns in the rebuttal. Therefore, I am raising my score to 4.

**Quality:**

2

**Strengths And Weaknesses:**

Strengths:
1: The paper addresses the issue of performance unfairness resulting from inconsistent client participation—a practical and important challenge in real-world federated learning.

2: The proposed method leverages the DEAL and ISPU modules to guide learning in a way that improves both local accuracy and performance fairness across clients.

3: The paper is well-written and easy to follow.

Weakness:

See the Question section.

---

> ### Author Rebuttal · Authors · 2025-07-30
>
> We sincerely thank you for your constructive and helpful comments. Below we address your concerns in order.
>
> ---
>
> ## Response to Question 1:
>
> Thank you for your meticulous observation. The statement in Line 27 refers specifically to the client local models $\boldsymbol{w}_i$, which are indeed heterogeneous and not assumed to be identical. In contrast, the small auxiliary model $\boldsymbol{g}_i$ in DEAL serves a shared, compact knowledge representation space, designed to facilitate efficient data-free distillation and representation alignment across diverse client models. Its homogeneity simplifies the distillation process by providing a stable common reference for knowledge transfer, without imposing any identical model constraints on the clients' local models.
>
> Relaxing the homogeneity assumption for the auxiliary model would introduce significant challenges:
>
> 1. **Architectural Mismatch:** In each round, the server aggregate the auxiliary models from all clients to synthesize a global model $\\mathcal{G}$. If the auxiliary models were architecturally heterogeneous, this aggregation would become infeasible, breaking the mechanism for integrating public information and reverting the problem to standard model-heterogeneous federated learning.
> 2. **Semantic Alignment:** Without a shared architectural basis, ensuring that feature representations from different auxiliary models are semantically aligned is extremely difficult. Any solution would likely require complex and costly techniques, potentially negating the efficiency of our method.
>
> Therefore, employing a homogeneous auxiliary model represents a practical and scalable design choice, effectively balancing efficient knowledge transfer with the inherent heterogeneity of client models.
>
> ---
>
> ## Response to Question 2:
>
> Thank you for your insightful comments. The appearance of slight fluctuations is primarily due to the fine granularity of the y-axis scale, which exaggerates minor variations. In reality, the values of $\lambda$ demonstrate substantial stability towards approximately 100 rounds. To validate this, we extended our experiments under the same setup as Figure 3c to 300 communication rounds. The results showed that the $\lambda$ values at round 300 differed by less than 0.01 compared to those at round 100 for both datasets. This stability aligns with the overall model convergence trends, as evident from the accuracy curves shown in Figure 6 and Figure 7. Due to the restrictions on image display here, we will adjust Figure 3 in the revised manuscript to more accurately illustrate the convergence behavior of $\lambda$.
>
> ---
>
> ## Response to Question 3:
>
> Regarding theoretical convergence, our framework builds upon established principles: prior studies have already proven the convergence of FL under unknown participation rates [1] and with model pruning mechanisms [2,3]. Our PHP-FL does not violate the convergence guarantees of FL. Specifically, the convergence of each local heterogeneous model $\boldsymbol{w}_i$ can be directly inferred from existing FL convergence analyses [4,5], as these models are trained locally with standard optimization procedures and the bias resulting from aggregation has been eliminated. For the homogeneous auxiliary model $\boldsymbol{g}_i$, its training with frozen local models is guaranteed to converge under challenges like submodel heterogeneity, non-uniform training, and data heterogeneity, as demonstrated in works like RAM-Fed [6], establishes a convergence rate of $O(1/\sqrt{T})$ in a comparable setting. The convergence of both the local and auxiliary models ensures that their ensemble output will reach a stable and consistent state. This conclusion is further supported by the empirical convergence behaviors demonstrated in experimental results.
>
> > [1] S, Wang, et al. A Lightweight Method for Tackling Unknown Participation Statistics in Federated Averagig. ICLR24.
> >
> > [2] H, Zhou et al. On the convergence of heterogeneous federated learning with arbitrary adaptive online model pruning. arXiv 2022.
> >
> > [3] J, Yuang, et al. Model pruning enables efficient federated learning on edge devices. TNNLS 2022.
> >
> > [4] L, Xiang, et al. On the convergence of fedavg on non-iid data. arXiv 2019.
> >
> > [5] L, Yi, et al. Federated model heterogeneous matryoshka representation learning. Neurips 2024.
> >
> > [6] Y, Wang, et al. Theoretical Convergence Guaranteed Resource-Adaptive Federated Learning with Mixed Heterogeneity. KDD 2023.
>
> ---
>
> ## Response to Minor Question 1:
>
> We appreciate the reviewer's insightful observation regarding Figure 1c and its connection to Definition 3. While Definition 3 defines client participation probabilities as linearly ordered, our experimental implementation (please refer to our anonymous code), introduces a permutation operation to $\\{p_i^t\\}_{i=1}^K$ to ensure randomness and eliminate any implicit ordering bias among clients. Consequently, Figure 1c visualizes the permuted order of these linearly distributed participation probabilities, which aligns with our experimental setup. To avoid potential misunderstanding, we have added a detailed explanation in the revised manuscript.
>
> ---
>
> ## Response to Minor Question 2:
>
> As explained in **Response to Minor Question M1**, although $p_i^t$ is initially defined as a function of client index $ i $ to establish the distribution of participation probabilities (e.g., *linear* pattern), our experimental implementation introduces a permutation operation on the set $\\{p_i^t\\}_{i=1}^K$. This permutation randomly reassigns the participation probabilities among clients, ensuring that the probability assigned to client $i$ is independent of the client index and other clients’ probabilities. This design eliminates cross-client dependencies, preserving fairness and randomness in practice. We have included a detailed explanation of this permutation mechanism in the revised manuscript to prevent potential misunderstandings.
>
> ---
>
> Overall, we hope that our responses can fully address your comments and will be grateful for any feedback.

---

> > ### Comment · Reviewer_VyUg · 2025-08-05
> >
> > Thank you for the detailed responses. The authors have thoroughly addressed my concerns in the rebuttal. Therefore, I am raising my score to 4.

---

> > > ### Author Response · Authors · 2025-08-06
> > >
> > > Dear Reviewer,
> > >
> > > We sincerely appreciate your valuable feedback and are glad that our revisions have addressed your concerns. Thank you for recognizing our efforts through the improved scores. We truly appreciate your constructive input throughout this process.
> > >
> > > Best, Authors

---

### Official Review · Reviewer_KfpB · 2025-06-15

**Clarity:** 3
**Significance:** 2
**Originality:** 2
**Rating:** 4
**Confidence:** 4

**Summary:**

This paper proposes an approach for model-heterogeneous federated learning, with an aim to improve the fairness among clients considering the probabilities of their participations. The main contributions consists of two parts, one is the auxiliary model to transfer the knowledge among clients and the second is a selective parameter update mechanism.

**Questions:**

1）Could using a fixed adaptability set to train the ensemble learning coefficients lead to overfitting, especially when these coefficients are retrained every round?
2）How does the model perform under varying degrees of data distribution heterogeneity across clients? While the datasets used in the study already exhibit some heterogeneity, it would be valuable to clarify how the model performs under different levels of heterogeneity.
3）In the experiments, are the performance metrics reported based on the local models of each client or the ensemble-averaged results after aggregation? This was not explicitly clarified in the text.
4）K is used to represent the total number of clients and a tunable sharpness hyperparameter. I suggest to use two symbols.

**Ethical Concerns:**

["NO or VERY MINOR ethics concerns only"]

**Final Justification:**

Considering the contributions and the rebuttal, I keep my score.

**Limitations:**

I think the limitations addressed by the author are appropriate.

**Paper Formatting Concerns:**

There is no formatting issues.

**Quality:**

2

**Strengths And Weaknesses:**

Strengths:
1) The paper is well-organized and the presentation is easy to follow.
 2) The designs of the two critical components have well-defined motivations. For the Dual-End Aligned Ensemble learning module, the integration of representation alignment and ensemble learning objectives facilitates both knowledge transfer and localized learning. For the Importance-Driven Selective Parameter Update module, the selection of important parameters for update aligns with the needs of adapting to varying participation frequencies among clients.
3）The experiments are comprehensive. The method was compared with multiple state-of-the-art heterogeneous federated learning approaches. Specifically, three participation probability models were simulated to validate the effectiveness of the proposed method under diverse scenarios. Additional experiments, including results under model-homogeneous settings, are provided in the appendix.

Weaknesses:
1）As shown in Section 5.2, while there are some improvements, the gains over existing methods are not particularly significant.
2）The method lacks theoretical analysis. For instance, there is no theoretical justification for why the selective parameter update mechanism adapts to different participation probability models. Intuitively, defining Equation (9) specifically for different participation probability models might yield better performance. Additionally, no theoretical analysis of convergence is provided.
3） The proposed approach incurs notable additional computational overhead.

---

> ### Author Rebuttal · Authors · 2025-07-30
>
> We sincerely thank you for your constructive and helpful comments. Below we address your concerns in order.
>
> ---
>
> ## Response to Weakness 1:
>
> We appreciate your observation regarding the performance gains. While some gains may appear modest, PHP-FL consistently achieves superior accuracy and fairness, especially under challenging conditions. For instance, on the Fashion-MNIST dataset, compared to the best baseline, PHP-FL achieves a 0.87% improvement in accuracy (AM) and a significant 2.48% reduction in the fairness metric (FM), indicating enhanced fairness. On the more complex CIFAR-10 dataset, the gains are even more substantial, with a 6.51% increase in AM and a 1.33% improvement in FM.
>
> More importantly, PHP-FL consistently achieves state-of-the-art performance for both accuracy and fairness across all evaluated scenarios. This dual enhancement, particularly in balancing these often conflicting objectives, marks a substantial advancement in heterogeneous federated learning, where enhancing one metric often compromises the other.
>
> ---
>
> ## Response to Weakness 2:
>
> Thank you for your insightful comments. Our selective parameter update mechanism (ISPU) is specially designed to be adaptive, focusing on task-relevant parameters while suppressing noisy or redundant updates. This design serves multiple purposes: it enables straggling clients to rapidly catch up upon rejoining training, maintains the learning momentum of active clients, mitigates conflicts arising from heterogeneous data, and ensures the retention of critical knowledge via high-importance parameters. Crucially, this mechanism operates without requiring prior knowledge of client participation patterns, which are inherently unpredictable in real-world scenarios. While some existing FL pruning methods pre-define pruning ratios [1] or adjust pruning ratios dynamically based on metrics like accuracy or training rounds [2, 3], our approach takes a pragmatic heuristic strategy. By estimating client lag from participation history to determine the update ratio $\alpha$, we address the uncertainty of client availability in a practical and effective manner. PHP-FL's superior performance, further validated under a realistic *Markovian* participation pattern ( please see **Response to Weakness 3** from **Reviewer zpY2**), confirms the robustness of this design.
>
> Regarding theoretical convergence, prior studies have already proven the convergence of FL under unknown participation rates [4] and with model pruning mechanisms [5]. Our PHP-FL does not violate the convergence guarantees of FL. Specifically, the convergence of each local heterogeneous model $\boldsymbol{w}_i$ can be directly inferred from existing FL convergence analyses, as these models are trained locally with standard optimization procedures and the bias resulting from aggregation has been eliminated. For the homogeneous auxiliary model $\boldsymbol{g}_i$, its training with frozen local models is also guaranteed to converge under challenges such as submodel heterogeneity, non-uniform training, and data heterogeneity, as demonstrated in works like RAM-Fed [6], which establishes a convergence rate of $O(1/\sqrt{T})$ in a comparable setting. The convergence of both the local and auxiliary models ensures that their ensemble outputs will reach a stable state. This conclusion is further supported by the empirical convergence behaviors observed in experimental results.
>
> > [1] Li, A, et al. Hermes: an efficient federated learning framework for heterogeneous mobile clients. MobiCom 2021.
> >
> > [2] L. Yi , et al. FedPE: Adaptive Model Pruning-Expanding for Federated Learning on Mobile Devices. TMC 2024.
> >
> > [3] Y. Jia, et al. DapperFL: domain adaptive federated learning with model fusion pruning for edge devices. NeurIPS 2024.
> >
> > [4] S. Wang, et al. A Lightweight Method for Tackling Unknown Participation Statistics in Federated Averaging. ICLR24.
> >
> > [5] H, Zhou, et al. On the convergence of heterogeneous federated learning with arbitrary adaptive online model pruning. arXiv 2022.
> >
> > [6] Y. Wang, et al. Theoretical Convergence Guaranteed Resource-Adaptive Federated Learning with Mixed Heterogeneity. KDD 2023.
>
> ---
>
> ## Response to Weakness 3:
>
> In PHP-FL, the main source of additional computation can be derived from training the homogeneous auxiliary model $\boldsymbol{g}_i^t$. However, compared to other auxiliary model-based methods like FedAPEN and FedMRL, our analysis in Table 10 of Appendix C.6 shows that PHP-FL incurs only a modest increase in FLOPs per communication round (5.54% and 7.40% respectively). Importantly, despite this slightly higher per-round computational cost, PHP-FL's faster convergence rate (as shown in Figure 6 and Figure 8 in Appendix) results in significantly lower overall computational requirements to reach the target performance. This highlights our method's efficiency in practical scenarios.
>
> We further evaluated the total costs for achieving 50% accuracy compared to non auxiliary model-based methods. As shown in Table R1, while non auxiliary baselines like FedGen and FedTGP exhibit lower per-round communication costs, our method PHP-FL, though incurring a higher communication cost per round due to auxiliary model transmission, achieves much faster convergence. This rapid convergence significantly reduces the overall computational costs, offering a considerable advantage in overall compute efficiency. Additionally, in practical deployments, the communication cost of PHP-FL can be further reduced by adopting smaller auxiliary models and applying bitwidth reduction techniques to the mask $M^t_i$.
>
> [**Table R1**: Comparison of total costs (communication rounds, computation cost, and communication cost) required to achieve 50% accuracy on CIFAR-10 under the *uniform* pattern. '/' indicates that the method failed to reach 50% accuracy.]
>
> |          Methods          | FML  | FedGen |  FedKD  | FedAPEN | FedTGP | FedMRL  | PHP-FL (Ours) |
> | :-----------------------: | :--: | :----: | :-----: | :-----: | :----: | :-----: | :-----------: |
> |   Communication Rounds    |  /   |   19   |   10    |    5    |   43   |    6    |       2       |
> | Computation Cost (TFLOPs) |  /   |  7.44  |  7.54   |  3.86   | 16.68  |  4.54   |     1.63      |
> |  Communication Cost (MB)  |  /   | 112.48 | 2002.60 | 1121.00 |  8.26  | 1344.30 |     230.8     |
>
> ---
>
> ## Response to Question 1:
>
> To specifically prevent such overfitting, we randomly hold out a tiny adaptability set $D_i^a$ from each client's local training set $D_i$ at the beginning of every communication round. This round-wise resampling of the adaptability set guarantees that the ensemble coefficient $\lambda_i$ is continuously optimized on fresh and unbiased data, thereby effectively mitigating overfitting risks. We have added a detailed explanation of this mechanism in the revised manuscript.
>
> ---
>
> ## Response to Question 2:
>
> We appreciate your constructive suggestion. To evaluate PHP-FL's robustness under varying data heterogeneity, we conducted additional experiments on CIFAR-10 using the Dirichlet distribution with $\beta=0.1$ (high heterogeneity) and $\beta=5$ (low heterogeneity). As shown in Table R2, PHP-FL consistently outperforms all baselines across both settings. Under high heterogeneity, PHP-FL surpasses the best-performing baseline by 2.09% in accuracy (AM) and reduces the fairness metric (FM) by 1.84%. This advantage becomes even more pronounced under low-heterogeneity, where PHP-FL achieves a remarkable 8.83% accuracy gain over the next best method while maintaining the best fairness performance. These results clearly show that PHP-FL is not only robust to different levels of data heterogeneity but consistently achieves state-of-the-art performance in both accuracy and fairness.
>
> [**Table R2**: Comparison results on CIFAR-10 under varying degrees of data distribution heterogeneity across clients. All other settings follow their default configurations.]
>
> | $\beta$ | Metrics  |   Standalone    |       FML       |     FedGen      |      FedKD      |     FedAPEN     |     FedTGP      |     FedMRL      |  PHP-FL (Ours)  |
> | :-----: | :------: | :-------------: | :-------------: | :-------------: | :-------------: | :-------------: | :-------------: | :-------------: | :-------------: |
> |  $0.1$  | AM (%) ↑ | $84.89\\pm0.07$ | $80.69\\pm0.16$ | $78.34\\pm0.37$ | $84.08\\pm0.04$ | $85.97\\pm0.12$ | $78.20\\pm0.49$ | $84.58\\pm0.14$ | $88.06\\pm0.04$ |
> |         | FM (%) ↓ | $14.57\\pm0.20$ | $18.01\\pm0.78$ | $20.93\\pm0.19$ | $14.94\\pm0.26$ | $13.58\\pm0.36$ | $22.93\\pm1.37$ | $15.66\\pm0.81$ | $11.74\\pm0.25$ |
> |   $5$   | AM (%) ↑ | $34.81\\pm0.10$ | $23.69\\pm0.25$ | $34.15\\pm0.36$ | $32.48\\pm0.31$ | $38.67\\pm0.31$ | $33.09\\pm0.30$ | $31.48\\pm0.24$ | $47.50\\pm0.43$ |
> |         | FM (%) ↓ | $4.89\\pm0.21$  | $6.19\\pm0.15$  | $4.67\\pm0.17$  | $4.54\\pm0.41$  | $5.43\\pm0.33$  | $4.37\\pm0.44$  | $8.29\\pm0.27$  | $4.21\\pm0.47$  |
>
> ---
>
> ## Response to Question 3:
>
> Thank you for the valuable question. As detailed in our methodology, each client $i$'s prediction is generated by the weighted model ensemble of its local model $\boldsymbol{w}_i$ and the auxiliary model $\boldsymbol{g}_i$ (Eq.8). Consequently, and in line with our Design Goals (Section 3), each client's test accuracy and fairness are computed from this ensemble output. The overall Accuracy Metric (AM) and Fairness Metric (FM) are reported as the average across all clients, ensuring the evaluation directly reflects the practical performance experienced by clients.
>
> ---
>
> ## Response to Question 4:
>
> We appreciate the your careful observation regarding the use of the symbol $K$. To resolve this ambiguity, we have revised the manuscript by using distinct symbols to separately denote the total number of clients and the sharpness hyperparameter.
>
> ---
>
> Overall, we hope that our responses can fully address your comments and will be grateful for any feedback.

---

> > ### Comment · Reviewer_KfpB · 2025-08-04
> >
> > Thank you for your response.

---

> > > ### Author Response · Authors · 2025-08-06
> > >
> > > Dear Reviewer,
> > >
> > > Thank you for your valuable feedback. We sincerely appreciate your time and constructive suggestions.
> > >
> > > Best, Authors

---

### Official Review · Reviewer_i57X · 2025-06-26

**Clarity:** 3
**Significance:** 2
**Originality:** 3
**Rating:** 4
**Confidence:** 4

**Summary:**

This work addresses performance degradation due to nonuniform participation of clients in FL and proposes a new FL framework
called PHP-FL in model heterogeneous FL environment that can improve model accuracy and performance fairness among clients.

**Questions:**

1. One main concern is about the communication and computation cost to get and train auxiliary models
for clients. The cost also depends on the size of auxiliary models.
In the appendix the PHP-FL is compared with other auxiliary model-based models from the
perspective of communication cost, but a comparison with non auxiliary model-based
models is necessary from the perspective of both costs.

2. The impact of masking needs to be investigated in detail. When the local dataset does not
contain full information on key features, masking based on the important parameters of
each local model, may miss some key features. One important reason for the use of a global
model is to utilize some missing key features, if any, for each local client. It seems that masking
may reduce the impact of the use of the global model.

3. In eq. 2, backbone alignment and predictor alignment are considered through L_{MMD} and
D_{KL}, respectively, but the scales of both terms may be different, so does it make sense to consider
the simple sum of both terms with different scales?
In eq. 3 how do we select the best for the customizable projection functions f and h?

4. In Fig. 2, $M^h_\*$ appears in the server, but no explanation is provided on it.
It seems to be a typo and hence is replaced by $M^h_i$ according to the expanation in section 4.1.
In step 6 of section 4.1, no explanation on how to update $M^h_i$ using $M^t_i$. Are they the same?

5. In $L^w_{ENS}$ (eq. 4) the local model w is used in the first term and the second term,
which seems to be overlapping. What happens if only the second term is considered?

6. What model architecture is used in the experiments for the global model? Is it large enough to have
full trained information in it?

7. In Fig. 3, it looks like that $\lambda$ does not converge until communication round 100.
There is a possibility that more communication rounds would improve the performance by choosing a better $\lambda$.
A discussion on the convergence of $\lambda$ would help understand the performance behavior.

8. Improvement on unfairness due to frequently and infrequent active participation:
The PHP-FL can improve the fairness in accuracy as shown in the experiment.
Since other auxiliary heterogeneous models also consider auxiliary models in training, what is the main
reason why the PHP-FL outperforms other models?

**Ethical Concerns:**

["NO or VERY MINOR ethics concerns only"]

**Final Justification:**

I have carefully reviewed the responses. Since my concerns have been addressed, I keep my score.

**Limitations:**

The proposed approach has relatively high communication and computational costs by considering auxiliary models
for clients.

**Paper Formatting Concerns:**

No paper formatting concerns are found.

**Quality:**

3

**Strengths And Weaknesses:**

The proposed model can achieve good accuracy and performance fairness among clients.
Each local client has an auxiliary model as well as its local model and trains both models,
which obviously increases computational cost and communication cost.
Masking information for each client is also uploaded, which also increases communication cost.

---

> ### Author Rebuttal · Authors · 2025-07-30
>
> We sincerely thank you for your constructive and helpful comments. Below we address your concerns in order.
>
> ---
>
> ## Response to Question 1:
>
> We sincerely appreciate the your valuable suggestion to compare our method with non auxiliary model-based approaches in terms of communication and computation costs. To address this, we have conducted additional experiments measuring the total costs (communication rounds, computation cost, and communication cost) required to achieve 50% accuracy. As shown in Table R1, while non auxiliary model-based baselines such as FedGen and FedTGP exhibit lower per-round communication costs (e.g., 112.48M and 8.26M respectively), our method PHP-FL, though incurring a higher communication cost per round due to auxiliary model transmission (230.8M), achieves significantly faster convergence (2 rounds versus 19 and 43 rounds, respectively). This rapid convergence significantly reduces the overall computational costs (1.63T for PHP-FL versus 7.44T and 16.68T respectively), offering a considerable advantage in overall compute efficiency. Moreover, in practical deployments, the communication cost of PHP-FL can be further optimized by adopting smaller auxiliary models and applying bitwidth reduction techniques to of the mask $M^t_i$.
>
> [**Table R1**: Comparison of total costs (communication rounds, computation cost, and communication cost) required to achieve 50% accuracy on CIFAR-10 under the *uniform* pattern in the heterogeneous setting. '/' indicates that the method failed to reach 50% accuracy.]
>
> |          Methods          | FML  | FedGen |  FedKD  | FedAPEN | FedTGP | FedMRL  | PHP-FL (Ours) |
> | :-----------------------: | :--: | :----: | :-----: | :-----: | :----: | :-----: | :-----------: |
> |   Communication Rounds    |  /   |   19   |   10    |    5    |   43   |    6    |       2       |
> | Computation Cost (TFLOPs) |  /   |  7.44  |  7.54   |  3.86   | 16.68  |  4.54   |     1.63      |
> |  Communication Cost (MB)  |  /   | 112.48 | 2002.60 | 1121.00 |  8.26  | 1344.30 |     230.8     |
>
>
> ---
>
> ## Response to Question 2:
>
> Our selective parameter update mechanism within the ISPU module is specifically designed to facilitate targeted selective fusion, rather than to block beneficial global knowledge. As defined in Eq.11, the update rule is given by: $\hat{\boldsymbol{g}}_i^{t-1} = \boldsymbol{g}_i^{t-1} \odot \neg M_i^h + \\mathcal{G}^{t-1} \odot M_i^h$, where the mask $M_i^h$ identifies the most signiﬁcant parameters of the local auxiliary model. Importantly, these significant parameters are directly updated by replacing their values with those from the global model $\\mathcal{G}^{t-1}$. This selective update mechanism ensures that clients can effectively acquire "missing key features" or knowledge embedded in the aggregated global model, which is particularly beneficial for straggling clients whose local parameters might be under-trained. Therefore, the masking mechanism does not diminish the global model's influence; rather, it adaptively transfers global knowledge where it is most needed via $M_i^h$, thereby achieving an effective fusion of global and local knowledge.
>
> ---
>
> ## Response to Question 3:
>
> Regarding the scales of $\mathcal{L}\_{MMD}$ and $\mathcal{D}\_{KL}$ in Eq. 2, both are fundamental divergence measures, and their composite sum is implicitly normalized by the mini-batch size. In practice, we observe that the joint optimization process naturally balances their contributions towards overall model alignment, as both terms implicitly guide gradient descent to minimize feature dissimilarities without requiring explicit weighting. Moreover, our MMD implementation further standardizes feature scales via internal normalization of representations (as seen in our anonymous code), which inherently aids their joint optimization.
>
> As for the projection functions $f$ and $h$ in Eq. 3, these are not externally defined hyperparameters. Instead, they represent learnable feature projection layers within the local model's backbone $\boldsymbol{w}_b$ and the auxiliary model's backbone $\boldsymbol{g}_b$, respectively. Their parameters are jointly learned during training, enabling adaptive projection of heterogeneous features into a shared latent space for effective alignment.
>
> ---
>
> ## Response to Question 4:
>
> We sincerely appreciate your meticulous observation. As described in step 6 of Section 4.1, $M_i^h$ represents the latest historical mask record of client $i$, which is maintained on the server. The update procedure is straightforward: for each active client $i$, the corresponding $M_i^h$ is directly replaced by the newly uploaded mask $M_i^t$ from the current communication round. Additionally, the notation $M^h_{* }$ in Figure 2 is not a typo. It denotes the entire collection of all clients' historical mask records maintained on the server, i.e., $\\{M\_i^h\\}\_{i=1}^K$ , distinguishing it from an individual client's mask $M_i^h$. To prevent further misunderstandings, we have provided a detailed clarification regarding both $M^h_*$ and the update procedure of $M_i^h$ in the revised manuscript.
>
> ---
>
> ## Response to Question 5:
>
> We thank the reviewer for this insightful question. This $\mathcal{L}_{ENS}^{\boldsymbol{w}}$ loss in Eq.4 comprises two cross-entropy terms, each serving a distinct purpose: the first term is designed to optimize the performance of local model $\boldsymbol{w}$ on its local data, while the second term focuses on improving the inference performance of the ensemble output (the inference process is shown in Eq.8). Maintaining the local model's performance via the first term is crucial, as a well-trained local model is essential for the subsequent knowledge alignment within our DEAL module. Removing this term would degrade the local model, thereby harming the alignment process and overall performance. We empirically validate this with an ablation study presented below. The results in Table R2 clearly show that omitting the first term leads to a significant performance drop across both datasets, e.g., a 2.73% decrease in accuracy on CIFAR-10 and a 0.47% decrease on Fashion-MNIST, with a corresponding decline in fairness. This confirms that both components are integral to our method's effectiveness.
>
> [**Table R2**: Ablation study on the first term of the $\mathcal{L}_{ENS}^{\boldsymbol{w}}$ loss function in Eq.4.]
>
> |   Datasets    | Metrics  | PHP-FL *w/o* first term of $ \mathcal{L}_{ENS}^{\boldsymbol{w}}$ |  PHP-FL (Ours)  |
> | :-----------: | :------: | :----------------------------------------------------------: | :-------------: |
> | Fashion-MNIST | AM (%) ↑ |                       $97.11\\pm0.12$                        | $97.58\\pm0.03$ |
> |               | FM (%) ↓ |                        $5.02\\pm0.37$                        | $4.29\\pm0.04$  |
> |   CIFAR-10    | AM (%) ↑ |                       $64.05\\pm0.20$                        | $66.78\\pm0.36$ |
> |               | FM (%) ↓ |                        $8.89\\pm0.18$                        | $8.09\\pm0.27$  |
>
>
>
> ---
>
> ## Response to Question 6:
>
> For a fair comparison, we consistently adopt GoogLeNet as the auxiliary model architecture for all methods, including our PHP-FL. GoogLeNet is selected because it has the smallest parameter count among the heterogeneous model group, while remaining sufficiently large to save comprehensive trained information, ensuring a robust and equitable evaluation across all auxiliary model-based methods.
>
> ---
>
> ## Response to Question 7:
>
> We appreciate the your insightful observation regarding the convergence of the average ensemble weight $\lambda$ in Figure 3. The appearance of slight fluctuations is primarily due to the fine granularity of the y-axis scale, which exaggerates minor variations. In reality, the values of $\lambda$ exhibit substantial stability towards approximately100 rounds. To validate this, we extended the experiments under the same setup as Figure 3c to 300 communication rounds. The results showed that $\lambda$ values at round 300 differed by less than 0.01 compared to those at round 100 for both datasets. This stability aligns with the overall model convergence, as evident from the accuracy curves in Figure 6 and Figure 7. Due to the restrictions on image display here, we will adjust Figure 3 in the revised manuscript to more accurately illustrate the convergence behavior of $\lambda$.
>
> ---
>
> ## Response to Question 8:
>
> While other auxiliary model-based heterogeneous FL methods also leverage auxiliary models, the distinct advantage of PHP-FL lies in its dual-module design, which is specifically tailored to address inconsistent client participation and model heterogeneity simultaneously.
>
> The DEAL module extends beyond conventional auxiliary model usage by integrating data-free knowledge distillation and representation alignment for robust dual-end knowledge transfer, enabling more effective alignment among diverse local models. Furthermore, the ISPU module directly tackles the fairness challenge arising from inconsistent participation. Unlike existing methods, it adaptively adjusts the update proportion $\alpha$ and selectively updates parameters based on each client’s participation history and parameter importance. This mechanism ensures that straggling clients receive prioritized global updates for faster catch-up, while frequently active clients retain their unique knowledge without being negatively influenced by lagging participants. This adaptive and importance-driven selection strategy, which is absent in other auxiliary model-based methods, empowers PHP-FL to achieve superior and more consistent fairness, alongside maintaining high accuracy.
>
> ---
>
> Overall, we hope that our responses can fully address your comments and will be grateful for any feedback.

---

> > ### Comment · Reviewer_i57X · 2025-08-04
> >
> > I thank the authors for their rebuttal. Since my questions have been addressed well, I will keep my score.

---

> > > ### Author Response · Authors · 2025-08-06
> > >
> > > Dear Reviewer,
> > >
> > > Thank you for your valuable feedback. We are pleased to have addressed your questions and greatly appreciate your time and constructive suggestions.
> > >
> > > Best, Authors

---

### Official Review · Reviewer_zpY2 · 2025-07-03

**Clarity:** 3
**Significance:** 3
**Originality:** 3
**Rating:** 4
**Confidence:** 3

**Summary:**

This paper addresses the underexplored challenge of performance fairness in model-heterogeneous federated learning (MH-FL) under non-uniform client participation probabilities. The authors propose PHP-FL, which includes two key modules: (1) the Dual-End Aligned ensemble Learning (DEAL) module for bidirectional alignment between local and auxiliary models using MMD and KL loss, and (2) the Importance-driven Selective Parameter Update (ISPU) module, which adaptively updates task-relevant parameters based on participation frequency. Extensive experiments on Fashion-MNIST and CIFAR-10 under three participation patterns demonstrate that PHP-FL achieves strong performance in both accuracy and fairness, outperforming prior MH-FL baselines.

**Questions:**

See weakness

**Ethical Concerns:**

["NO or VERY MINOR ethics concerns only"]

**Final Justification:**

Thank for authors' rebuttal, which addressed my concerns. I will maintain my positive attitude and score.

**Limitations:**

Yes (clearly acknowledged in Section 6 and the experiments)

**Paper Formatting Concerns:**

Conforms to NeurIPS formatting. Figures and tables are legible.

**Quality:**

3

**Strengths And Weaknesses:**

**Strengths**:

The paper tackles a timely and practical problem — performance unfairness under inconsistent participation in MH-FL — which has been insufficiently explored in prior work. This brings new perspectives to the FL fairness literature.

The DEAL module incorporates both representation-level (via MMD loss) and prediction-level (via KL divergence) alignment, and the ISPU module introduces a sigmoid-based update ratio to balance frequent and infrequent clients, both of which are thoughtfully designed and motivated.

PHP-FL consistently outperforms state-of-the-art MH-FL baselines (FedAPEN, FedMRL, etc.) across multiple participation patterns and datasets, with improvements in both average accuracy and fairness. The ablation study validates the complementary effects of DEAL and ISPU.

The methodology is clearly presented with good visual illustrations (e.g., Fig. 2, Fig. 4), and code is released anonymously, enabling reproducibility.

**Weaknesses**:

- The experiments are only conducted on two relatively simple vision datasets (Fashion-MNIST and CIFAR-10). These may not sufficiently reflect the complexity and scale of real-world MH-FL applications. Adding results on at least one larger-scale or cross-domain dataset (e.g., CelebA, Reddit, or MIMIC-III) would significantly strengthen the empirical evaluation.


- The proposed method introduces multiple additional components (auxiliary model training, MMD computation, adaptive masking), which may increase training cost and memory usage. However, there is no analysis of overhead (e.g., training time, FLOPs, memory footprint), nor comparison with lightweight alternatives. This may hinder practical deployment.

- The paper designs three participation patterns (uniform, normal, linear), but these are synthetic and static. In real-world scenarios, client availability is often temporally correlated or bursty. The current evaluation may not reflect PHP-FL’s robustness under such dynamic conditions.

---

> ### Author Rebuttal · Authors · 2025-07-30
>
> We sincerely thank you for your constructive and helpful comments. Below we address your concerns in order.
>
> ---
>
> ## Response to Weakness 1:
>
> Thanks for the valuable comments. To address this concern, we have conducted additional experiments on CelebA dataset to further evaluate the generalizability of PHP-FL.
>
> From Table R1, we still see that the results on CelebA further validate the effectiveness and robustness of PHP-FL, demonstrating consistently superior performance in both accuracy and fairness, even with increased data complexity.
>
> [**Table R1**: Comparison results on CelebA dataset. The target attribute is *Smiling* and all other settings follow their default configurations.]
>
> |     Methods     |     Uniform     |     Uniform     |     Normal      |     Normal      |     Linear      |     Linear      |
> | :-------------: | :-------------: | :-------------: | :-------------: | :-------------: | :-------------: | :-------------: |
> |                 |    AM (%) ↑     |    FM (%) ↓     |    AM (%) ↑     |    FM (%) ↓     |    AM (%) ↑     |    FM (%) ↓     |
> | FedAPEN [KDD23] | $92.20\\pm0.14$ | $12.30\\pm0.76$ | $92.16\\pm0.10$ | $12.04\\pm0.43$ | $92.14\\pm0.08$ | $12.60\\pm1.17$ |
> | FedMRL [NIPS24] | $87.10\\pm0.67$ | $15.38\\pm0.72$ | $87.13\\pm0.70$ | $15.58\\pm0.50$ | $87.34\\pm0.99$ | $15.86\\pm0.37$ |
> |  PHP-FL (Ours)  | $93.43\\pm0.50$ | $11.68\\pm0.59$ | $93.34\\pm0.41$ | $11.89\\pm0.50$ | $93.41\\pm0.48$ | $12.19\\pm0.65$ |
>
>
> ---
>
> ## Response to Weakness 2:
>
> We fully agree that a thorough analysis of training cost and memory usage is essential for evaluating practical deployability. We would like to emphasize that a comprehensive analysis addressing these concerns has already been included in the Appendix. Specifically:
>
> 1. **Memory Footprint:** Table 5 in Appendix B.3 provides the parameter counts of all evaluated models, providing a clear comparison of their memory requirements.
> 2. **Communication and Computation Costs:** Appendix C.6 provides an in-depth analysis of the communication and computation costs, comparing PHP-FL with two state-of-the-art auxiliary model-based baselines. Our experimental results, presented in Figure 8, demonstrate that while PHP-FL incurs slightly higher per-round communication and computation costs, its significantly faster convergence rate ultimately results in much lower overall resource consumption.
>
> In addition, we conducted further experiments to measure the total costs (communication rounds, computation cost, and communication cost) required to reach 50% accuracy compared to non auxiliary model-based methods. As shown in Table R2, while PHP-FL's per-round communication cost is slightly higher than non auxiliary model-based methods (e.g., FedGen and FedTGP) due to the transmission of auxiliary model, it reaches target accuracy in just 2 rounds, whereas all baselines require at least 5 rounds . Consequently, the total communication and computation costs are substantially lower. Furthermore, the per-round costs can be readily optimized in practice by employing smaller auxiliary models or mask quantization (e.g., bitwidth reduction). Thus, PHP-FL offers a far more efficient path to convergence in practical deployments.
>
> [**Table R2**: Comparison of total costs (communication rounds, computation cost, and communication cost) required to achieve 50% accuracy on CIFAR-10 under the *uniform* pattern in the heterogeneous setting. '/' indicates that the method failed to reach 50% accuracy.]
>
> |          Methods          | FML  | FedGen |  FedKD  | FedAPEN | FedTGP | FedMRL  | PHP-FL (Ours) |
> | :-----------------------: | :--: | :----: | :-----: | :-----: | :----: | :-----: | :-----------: |
> |   Communication Rounds    |  /   |   19   |   10    |    5    |   43   |    6    |       2       |
> | Computation Cost (TFLOPs) |  /   |  7.44  |  7.54   |  3.86   | 16.68  |  4.54   |     1.63      |
> |  Communication Cost (MB)  |  /   | 112.48 | 2002.60 | 1121.00 |  8.26  | 1344.30 |     230.8     |
>
> ---
>
> ## Response to Weakness 3:
>
> We appreciate your constructive suggestion. To address this valuable feedback, we have conducted additional experiments under the *Markovian* participation pattern following FedAU [1], which better reflect dynamic client availability in real-word scenarios. In this pattern, each client follows a two-state Markov chain to determine its participation status in each training round, where the two states correspond to "participating" and "not participating" in the current training round. This modeling introduces temporal correlation in client participation behavior while maintaining sufficient randomness, offering a more realistic simulation compared to independently sampled participation patterns.
>
> For the parameter settings of the *Markovian* pattern experiment, we constrain the maximum transition probability from the non-participating state to the participating state to 0.05, thereby avoiding excessively frequent state changes and ensuring realistic participation dynamics. The initial state of each client's Markov chain is randomly sampled according to the stationary probability, which is set to 0.5 to ensure that approximately half of the clients participate in each round on average. The transition probabilities are carefully calibrated to maintain this stationary distribution throughout the training process, ensuring system stability while introducing participation heterogeneity across clients.
>
> As shown in Table R3, the experimental results under the *Markovian* participation pattern further validate PHP-FL’s robustness, consistently achieving superior accuracy and fairness despite the increased dynamic and unpredictable client availability.
>
> [**Table R3:** Comparison results on CIFAR-10 dataset under the *Markovian* participation pattern.  All other settings follow their default configurations.]
>
> | Methods  |   Standalone    |       FML       |     FedGen      |      FedKD      |     FedAPEN     |     FedTGP      |     FedMRL      |  PHP-FL (Ours)  |
> | :------: | :-------------: | :-------------: | :-------------: | :-------------: | :-------------: | :-------------: | :-------------: | :-------------: |
> | AM (%) ↑ | $52.77\\pm1.44$ | $42.73\\pm0.96$ | $51.93\\pm0.84$ | $51.31\\pm0.80$ | $57.84\\pm0.53$ | $50.55\\pm1.79$ | $49.02\\pm1.74$ | $64.04\\pm0.73$ |
> | FM (%) ↓ | $16.10\\pm2.50$ | $19.29\\pm2.04$ | $14.91\\pm0.38$ | $16.82\\pm1.45$ | $13.99\\pm3.34$ | $12.14\\pm1.66$ | $15.86\\pm0.08$ | $11.69\\pm1.35$ |
>
>
> > [1] S. Wang and M. Ji. A Lightweight Method for Tackling Unknown Participation Statistics in Federated Averaging. ICLR 2024.
>
> ---
>
> Overall, we hope that our responses can fully address your comments and will be grateful for any feedback.

---

> > ### Comment · Reviewer_zpY2 · 2025-08-06
> > **Official comments by reviewer zpY2**
> >
> > Thank you to the authors for the rebuttal, which addressed my concerns. Therefore, I will keep my score unchanged.

---

> > > ### Author Response · Authors · 2025-08-07
> > >
> > > Dear Reviewer,
> > >
> > > Thank you for your valuable feedback. We are pleased to have addressed your concerns and greatly appreciate your time and constructive suggestions.
> > >
> > > Best, Authors

---

### Decision · Program_Chairs · 2025-09-17

**Decision:**

Accept (poster)

**Comment:**

This paper studies fairness in model-heterogeneous federated learning under non-uniform client participation. The proposed framework, PHP-FL combines DEAL for dual-end alignment and ISPU for adaptive parameter updates, aiming to improve both accuracy and fairness across clients. The method is well-motivated and shows consistent improvements over baselines on Fashion-MNIST and CIFAR-10. The main concerns are that experiments are limited to relatively simple datasets and the additional overhead introduced by auxiliary models and masking is not analyzed. Nevertheless, the contribution is clear, the design is sound, and the results are convincing. All reviewers give weak accept. I recommend acceptance as a poster. I also expect the authors can address all concerns of reviewers according to rebuttal in the camera ready version.